# WET: Mitigating World-Conditioned Knowledge Conflicts via World Entropy Tethering

**Zixuan Wang**[1][*]  **Yifei He**[1][*]  **Zihan Wang**[1]  **Kun Wang**[2]  **Chaomeng Chen**[3][4]

## Abstract

Large language models (LLMs) face a "loyalty dilemma" when correctness is conditioned on an active world-of-discourse. We identify a systemic failure mode—*world misattribution*—where models implicitly ground generation in an incompatible regime and drift from the target world. We propose **World Entropy Tethering (WET)**, an inference-time monitor-and-tether: a world-entropy probe flags drift risk on prompt anchors, and a conditional score matching geometry model identifies tethering heads for entropy-gated rescaling. Experiments show: (I) **Linear Separability**: world labels are linearly decodable from internal states; (II) **Geometric Drift**: hallucinations are preceded by measurable deviations from the target world region; and (III) **Targeted Mitigation**: WET improves world consistency and reduces hallucination rates by up to 22.4% without compromising generation quality. Code is available at https://github.com/guess-guess-who-i-am/WET-World-Entropy-Tethering.

## 1. Introduction

Large language models (LLMs) routinely face a *loyalty dilemma*: at inference time they must arbitrate between competing information sources—most notably the vast parametric knowledge encoded in their weights and the specific, sometimes contradictory evidence provided in the prompt or retrieved context (Liu et al., 2024; Shi et al., 2024b). Ideally, the model should remain faithful to a *target world-of-discourse*, a regime where the prompt's local facts and rules dominate over generic priors. We focus on settings

[*]Equal contribution [1]Harbin Institute of Technology, Harbin, China [2]Nanyang Technological University, Singapore [3]Great Bay University, Dongguan, China [4]Tsinghua University, Beijing, China. Correspondence to: Chaomeng Chen <cmchen@gbu.edu.cn>.

*Proceedings of the 43rd International Conference on Machine Learning*, Seoul, South Korea. PMLR 306, 2026. Copyright 2026 by the author(s).

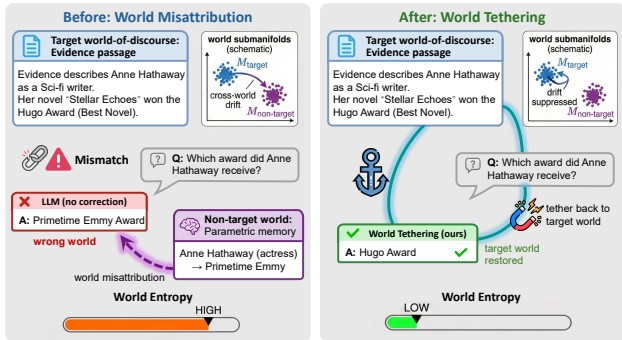

*Figure 1.* **World Misattribution as Geometric Drift.** (Left) Under knowledge conflicts, generation can drift from the target world-of-discourse toward incompatible priors or distractor contexts, which is foreshadowed by high *World Entropy* on prompt anchors. (Right) WET detects this instability and *tethers* the hidden-state trajectory back toward the target world region through a **dynamic, manifold-aware** and sparse, entropy-gated head intervention, reducing hallucinations and improving world consistency.

where the prompt explicitly specifies such a target world, including retrieved evidence sets in RAG, predefined fictional canons, or simulator specifications, and the desired output must remain consistent with that world. In practice, under knowledge conflicts, LLMs often fail to maintain this regime and drift toward incompatible priors or distractor narratives (Xie et al., 2024; Zhao et al., 2024), undermining reliability in evidence-based QA, RAG, and agentic simulations.

Existing approaches predominantly conceptualize conflicts as token- or proposition-level competition (e.g., selecting between two factual claims) and resolve them via direct inference-time steering, often by manipulating heads or suppressing components from conflicting sources (Li et al., 2025; Jin et al., 2024; Shi et al., 2024a). While effective in targeted cases, these methods typically treat failures as *local* discrepancies and rely on heuristics that do not explicitly model the *global* stability of the generation trajectory: a model may appear locally aligned early in the prompt yet later "slip" into an incompatible discourse regime. In contrast, we argue that many hallucinations and inconsistencies reflect a deeper systemic error: **world misattribution**. When a model hallucinates in RAG (Figure 1), it does not merely select an incorrect fact; it implicitly attributes generation to an incompatible regime, reminiscent of source-

monitoring errors in human cognition (Johnson et al., 1993).

We hypothesize that world misattribution has a concrete geometric signature: **Cross-World Drift** in the model's hidden-state space. Specifically, prompts from distinct worlds-of-discourse induce separable (often multi-modal) *regions* of activations. Under conflict, the hidden-state trajectory destabilizes and deviates away from the target world region toward a distractor region, and downstream generations inherit that drift. This geometric framing unifies diverse conflicts—parametric-memory intrusion, distractor contexts, competing narratives—under one principle: maintaining consistency is equivalent to keeping the trajectory within (or near) the target world region.

Guided by this perspective, we introduce **World Entropy Tethering (WET)**, a lightweight framework that *detects* incipient drift and *tethers* the trajectory back toward the target world without retraining the base LLM. WET leverages two complementary signals derived from internal activations. First, **World Entropy** is computed by a small world-posterior probe trained on a calibration set with discrete world labels; its entropy on prompt anchor states provides a robust *drift-risk* indicator by capturing uncertainty across competing worlds. In contrast, max-probability confidence can remain spuriously high under miscalibration or distribution shifts, masking instability until it is too late. Second, **World Geometry** is learned by estimating a world-conditioned vector field in activation space via conditional denoising score matching: the resulting score function provides a local direction that points toward high-density regions of the target world, explicitly modeling multi-modality where simple centroids are insufficient. Importantly, we train a small conditional score model *offline* on anchor states, avoiding any base-model finetuning.

Crucially, WET uses the learned geometry to identify a sparse set of **tethering heads** that are *mechanistically* critical for preserving world structure. We conduct a geometric sensitivity analysis: for each attention head, we counterfactually patch its anchor activation to a within-world mean and measure how much this ablation perturbs the target world score field in a noised space, yielding a robust, geometry-aware importance score. At inference time, WET adopts a minimal-intrusion policy: it continuously monitors Anchor Entropy and *only when* risk is high does it intervene by rescaling the identified tethering heads through a smooth gate, nudging the residual stream back toward the target world direction while preserving fluent generation when the model is already stable.

Our core contributions are summarized below:

- We propose **world misattribution** as a unifying account of knowledge conflicts, framing hallucinations and inconsistencies as *geometric drift* between discourse regimes

rather than isolated token-level errors.
- We introduce WET, a **monitor-and-tether** framework that combines **World Entropy** as an early drift-risk signal with **World Geometry** learned via conditional score matching on anchor states to obtain a local tethering direction.
- We develop a **geometric sensitivity** criterion to identify sparse **tethering heads** and an **entropy-gated** inference-time intervention that improves world consistency (up to 22.4%) while preserving generation quality across factual RAG and fictional consistency benchmarks.

## 2. Related Work

**Knowledge conflicts and RAG faithfulness.** Knowledge conflicts arise when parametric memory, contextual evidence, or multiple retrieved sources provide inconsistent signals (Xu et al., 2024; Wang et al., 2023; Liu et al., 2024). Benchmarks such as ConflictBank explicitly construct conflict instances (Su et al., 2024b), and RAG settings further surface conflicts due to ambiguity or contradictory evidence (Pham et al., 2024; Hou et al., 2024; Marjanović et al., 2024). A parallel line improves context-faithful RAG via self-reflection and refinement (Asai et al., 2024; Wu et al., 2025b; Shi et al., 2024b) or explicit conflict modeling (Zhang et al., 2025), often with minimal extra supervision.

**World consistency and role-play.** Fictional QA and role-play expose a structured failure mode: outputs can be coherent yet violate the active world's rules/canon (Ahn et al., 2024; Tang et al., 2025; Sadeq et al., 2024; Cabello et al., 2023). We connect these settings to knowledge conflicts by treating the world-of-discourse as the latent source variable that must remain stable throughout decoding.

**Probing and internal monitoring.** Uncertainty and internal-state signals are widely used to detect unreliable generation, including semantic uncertainty/entropy (Kuhn et al., 2023; Farquhar et al., 2024) and probe-based approximations from a single forward pass (Kossen et al., 2024; Chen et al., 2024; Beigi et al., 2024; Su et al., 2024a). Recent entropy-guided methods further use cross-layer token entropy for factual decoding, next-token entropy spikes for multi-turn resetting, or hidden-state entropy for efficient block pruning (Wu et al., 2025a; Mohammad Khalid et al., 2025; Yang et al., 2025). Additionally, identifying truth directions in activation space aids in directly monitoring factuality (Azaria & Mitchell, 2023; Li et al., 2023; Burns et al., 2022). Our monitor targets contextual world misattribution rather than factuality alone: it measures uncertainty over discourse regimes at prompt anchors and couples this signal to geometry-selected intervention heads.

**Plug-and-play inference-time control.** Mechanistic interventions can mitigate knowledge conflict by editing at-

tention heads or neurons at test time: PH3 prunes conflict-mediating heads (Jin et al., 2024), IRCAN reweights context-sensitive neurons (Shi et al., 2024a), and JUICE rescales influential heads via a dual-run scheme (Li et al., 2025). More generally, inference-time activation interventions/steering efficiently provide low-cost behavior control (Li et al., 2023; Hościłowicz et al., 2024; Zou et al., 2023; Chuang et al., 2024; Heimersheim & Nanda, 2024). WET follows this plug-and-play paradigm but adds a world-entropy monitor and a novel world-geometry signal for selecting and scaling a small set of world-critical heads.

## 3. Problem Setup

We consider an autoregressive language model $p_\theta$ that maps an input prompt $x$ to a response $y = (y_1, \ldots, y_T)$ via $y_t \sim p_\theta(\cdot \mid x, y_{<t})$. We analyze the model's internal representations via its residual stream: let $h_\ell^{(t)} \in \mathbb{R}^d$ denote the activation vector at layer $\ell$ and decoding step $t$. During the parallel prefix pass, we denote the activation at layer $\ell$ and prefix token index $j$ as $h_\ell[j](x) \in \mathbb{R}^d$. We use $h(x)$ to denote representations computed during the parallel processing of the prefix $x$, and $h(x, y_{<t})$ for representations generated dynamically during decoding.

### 3.1. World-Conditioned Generation

In grounded generation tasks (e.g., RAG, constrained reasoning), the validity of a generated response is not absolute but relative to a specific world-of-discourse. We formalize the discourse regime as a discrete latent variable $w \in \mathcal{W}$. In practice, $w$ selectively indexes the underlying authoritative source that should dominate decoding, such as a retrieved evidence set ($w_{evid}$), a specific fictional canon, or the model's internal parametric memory ($w_{param}$).

For each input $x$, we assume there exists a system-defined **target world** $w^\star \in \mathcal{W}$ (e.g., specified by the retrieved context). A strictly correct generation should remain consistent with $w^\star$ throughout the trajectory. To monitor this consistency, we assume the existence of a linear probe $q_\phi(w \mid h)$ trained on internal activations to approximate the posterior probability of the active world:

$$q_\phi(w \mid h) \approx \Pr(w \mid h; \theta). \quad (1)$$

Based on this probe, we define the **World Entropy** at any given intermediate hidden state $h$ as $H(h) = -\sum_{w \in \mathcal{W}} q_\phi(w|h) \log q_\phi(w|h)$, serving as a proxy for the model's uncertainty regarding the active discourse regime.

### 3.2. Formulating World Misattribution

A key insight of this work is that knowledge conflicts are not merely token-level errors but represent a trajectory-level deviation from the target world. We formalize this via **Anchor**

**States** and **Representation Drift**.

**Definition 3.1** (Prefix Anchor States). To enable lightweight monitoring without overhead during decoding, we designate a specific probe layer $\ell^*$. We define an anchor extraction operator $\mathcal{A}(\cdot)$ that selects hidden states from the prefix forward pass at pre-specified critical token indices $\mathcal{P} = \{p_1, \ldots, p_M\}$ (e.g., end-of-evidence tokens):

$$\mathcal{A}(x) = \{h_{\ell^*}[p_m](x) \mid m = 1, \ldots, M\}. \quad (2)$$

We define the **Anchor Entropy** as $\bar{H}(x) = \max_m H(h_{\ell^*}[p_m](x))$, which serves as a scalar metric of the model's initial confusion.

Leveraging these theoretical constructs, we rigorously formalize the failure mode central to this study.

**Definition 3.2** (Cross-World Drift). Let $w^\star$ be the target world. We say a generation $y$ exhibits Cross-World Drift if the internal states trajectory at probe $\{h_{\ell^*}^{(t)}\}_{t=1}^T$ satisfies:

$$\mathbb{E}_t\big[q_\phi(w^\star \mid h_{\ell^*}^{(t)})\big] < \delta, \quad \text{for some threshold } \delta, \quad (3)$$

implying the representation has migrated from the target manifold to a competing distractor regime (e.g., $q_\phi(w_{param} \mid h) \gg q_\phi(w^\star \mid h)$).

Our method relies on the premise that the stability of the autoregressive trajectory is correlated with the crispness of the world representation in the prefix.

**Assumption 3.3** (Anchoring Hypothesis). The risk of Cross-World Drift during decoding is monotonically increasing with the Anchor Entropy $\bar{H}(x)$. That is, initial confusion in the world-of-discourse propagates into trajectory instability.

Finally, we formally frame our proposed intervention goal as a standard constrained optimization problem.

> **Problem 3.4** (World Consistency Control). Given a prompt $x$ with target world $w^\star$, find an intervention function $g \in \mathcal{G}$ (applied to internal activations) that minimizes deviation from the base policy while satisfying a world-consistency constraint:
>
> $$\min_{g \in \mathcal{G}} \quad \mathbb{E}\big[\mathrm{KL}\big(p_{\theta,g}(\cdot \mid x) \,\|\, p_\theta(\cdot \mid x)\big)\big] \quad (4)$$
>
> $$\text{s.t.} \quad \Pr_{y \sim p_{\theta,g}(\cdot|x)}\Big[y \text{ is consistent with } w^\star\Big] \geq 1 - \epsilon.$$

## 4. Methodology

We propose **World Entropy Tethering (WET)**, a lightweight framework designed to mitigate representation drift by anchoring the generation trajectory to the target discourse regime. WET operates in two phases: an offline

stage that learns the topological structure of world manifolds and identifies "tethering heads," and an online stage that dynamically intervenes based on prefix uncertainty.

## 4.1. Preliminaries and Notation

We consider an autoregressive transformer where the hidden state $h_\ell^{(t)} \in \mathbb{R}^d$ at layer $\ell$ and step $t$ is updated via residual connections. Let $\mathcal{D} = \{(x^{(j)}, w^{(j)})\}_{j=1}^N$ be a calibration dataset with discrete world labels $w \in \mathcal{W}$. Each input $x$ induces a set of prefix anchor states $\mathcal{A}(x) = \{h_{\ell^*}[p_m](x)\}_{m=1}^M$ extracted at pre-specified critical token indices (e.g., end-of-evidence). When discussing the intervention mechanism, we specifically focus on the tethering anchor $a = (p_a, \ell^*)$ (typically the last token of the prompt), denoting its state as $h_a(x)$.

## 4.2. Phase I: Learning World Geometry

To effectively tether the model, we first need a geometric signal indicating the local direction toward the target world manifold. Since world distributions can be multi-modal, simple global centroids are insufficient. We instead learn a world-conditioned vector field.

**World-Entropy Monitor.** We first train a lightweight linear probe $q_\phi(w \mid h)$ to estimate the posterior probability of the active world. We minimize the cross-entropy on the extracted anchor states:

$$\mathcal{L}_{\text{probe}} = -\mathbb{E}_{(x,w)\sim\mathcal{D}}\Big[\frac{1}{M}\sum_{m=1}^M \log q_\phi\big(w \mid h_{\ell^*}[p_m](x)\big)\Big]. \tag{5}$$

At inference time, the entropy of this probe, $H(h) = -\sum_w q_\phi(w|h) \log q_\phi(w|h)$, serves as a **drift-risk signal**. We posit that while confidence ($\max_w q_\phi$) can be spuriously high under distribution shifts, entropy robustly captures the model's confusion among competing discourse regimes.

**Manifold Geometry via Conditional Score Matching.** We model the gradient field of the world density using Conditional Denoising Score Matching (DSM). For a clean anchor state $h = h_a(x)$ associated with world $w$, we sample a noise level $u$ and perturb it with Gaussian noise:

$$\tilde{h} = h + \sigma_u\xi, \qquad \xi \sim \mathcal{N}(0, I). \tag{6}$$

We then train a score network $\hat{\xi}_\psi(\tilde{h}, u, w)$ to reconstruct the noise $\xi$ by minimizing:

$$\mathcal{L}_{\text{score}} = \mathbb{E}_{(x,w)\sim\mathcal{D}}\mathbb{E}_{u,\xi}\Big[\big\|\hat{\xi}_\psi(\tilde{h}, u, w) - \xi\big\|_2^2\Big]. \tag{7}$$

The optimal network provides an estimate of the conditional score function (the gradient of the log-density):

$$s_\psi(\tilde{h}, u, w) \triangleq \nabla_{\tilde{h}} \log p_u(\tilde{h} \mid w) \approx -\frac{1}{\sigma_u}\hat{\xi}_\psi(\tilde{h}, u, w). \tag{8}$$

Crucially, $s_\psi$ provides a vector locally pointing toward high-density manifold regions of world $w$, allowing us to quantitatively measure how specific model components contribute to maintaining this directional alignment.

## 4.3. Phase II: Identifying Tethering Heads

We seek to identify a sparse set of attention heads that causally shape the world manifold. We quantify this influence via a geometric sensitivity analysis.

**Counterfactual Patching Operator.** Let $o_{\ell,i}[p](x)$ denote the output of head $i$ at layer $\ell$ and position $p$. We first compute the within-world mean activation for each head at the anchor position:

$$\mu_w^{(\ell,i)} \triangleq \mathbb{E}_{x'\sim\mathcal{D}_w}\Big[o_{\ell,i}[p_a](x')\Big]. \tag{9}$$

We define a patching operator $\Pi_w^{(\ell,i)}$ that replaces the specific head output with this generic mean. The resulting perturbed anchor state consequently becomes:

$$\Pi_w^{(\ell,i)}\big(h_a(x)\big) \triangleq h_a\big(x;\, o_{\ell,i}[p_a] \leftarrow \mu_w^{(\ell,i)}\big). \tag{10}$$

This operation effectively "ablates" the local instance-specific world information carried by that head, replacing it with the global world vector average.

**Geometric Sensitivity Score.** We measure the impact of this ablation on the score field. To ensure robustness, we evaluate this in the noised space. Let $\tilde{h}$ be the noisy anchor and $\widetilde{\Pi} = \Pi_w^{(\ell,i)}(h_a(x)) + \sigma_u\xi$ be the noisy patched anchor (using the same $\xi$). **Geometric Sensitivity** is defined as:

$$S_w(\ell, i) = \mathbb{E}\Big[\big\|s_\psi(\tilde{h}, u, w) - s_\psi(\widetilde{\Pi}, u, w)\big\|_2\Big]. \tag{11}$$

**Rationale: Isotropic Norm vs. Signed Projection.** A key design choice in Eq. (11) is the use of the $\ell_2$ norm rather than a signed projection (e.g., inner product with the gradient direction). We justify this choice based on the high-dimensional nature of the representation space:

- **Signal Cancellation:** Active heads often exhibit fluctuating directional alignment with the local gradients across diverse prompts. A signed summation can cause these contributions to cancel out (i.e., $\sum(+\delta) + (-\delta) \approx 0$), leading to the underestimation of heads that are highly influential but directionally complex.

- **Robustness:** The magnitude of the gradient shift captures the geometric causality of a head. Our usage of the norm ensures we select heads that strictly maintain the manifold structure, regardless of the local curvature direction.

Finally, we normalize these scores layer-wise to $\bar{S}_w(\ell, i)$ to handle scale differences across depths.

### 4.4. Phase III: Inference-Time Tethering

During inference, we apply an entropy-gated intervention that amplifies the tethering heads only when necessary.

**Dynamic Intervention Mechanism.** Given a test input $x$ and target world $w^\star$, we first compute the Anchor Entropy $\bar{H}(x)$. We define a continuous gating factor $\alpha(x) \in [0, 1]$:

$$\alpha(x) = \left[ \frac{\bar{H}(x) - \tau}{\log |\mathcal{W}| - \tau + \eta} \right]_0^1, \qquad (12)$$

where $\tau$ is a validation-tuned threshold. Let $h_\ell^{(t)}$ denote the residual stream at decoding step $t$. The standard update is $h_{\ell+1}^{(t)} = h_\ell^{(t)} + \sum_i o_{\ell,i}^{(t)} + \text{MLP}(h)$. We modify this update by rescaling the top-$K$ sensitive heads $\mathcal{K}_{w^\star}$:

$$h_{\ell+1}^{(t)} = h_\ell^{(t)} + \sum_i \gamma_{\ell,i} \cdot o_{\ell,i}^{(t)} + \text{MLP}(h), \qquad (13)$$

where the scaling factor $\gamma_{\ell,i}$ is defined as:

$$\gamma_{\ell,i} = \begin{cases} 1 + \alpha(x) \cdot \lambda \cdot \max(0, \bar{S}_{w^\star}(\ell, i)) & \text{if } (\ell, i) \in \mathcal{K}_{w^\star}, \\ 1 & \text{otherwise.} \end{cases} \qquad (14)$$

This mechanism ensures a minimal-intrusion policy: when the prompt anchor is well-aligned with the target world (low entropy), $\alpha \approx 0$ and the model behaves normally. Only when the anchor indicates ambiguity does WET tether the trajectory by amplifying the world-critical heads.

## 5. Experimental Setup

As summarized in Figure 2, we systematically evaluate WET within a unified world-conditioned conflict framework. Our experimental design assesses two complementary dimensions: **(i) Response Fidelity**, and **(ii) World Adherence**. all main results are reported as the mean $\pm$ standard deviation across 3 independent runs. Detailed hyperparameter settings are available in Appendix C.

### 5.1. Datasets and Protocol

**Standardized Prompting and Anchoring.** All inputs are formatted into three explicit blocks: [SPEC] (declares the target world $w^\star$), [EVIDENCE] (supporting or conflicting context), and [QUESTION] (the user query). Each block ends with a dedicated marker token, from which we extract

*prefix anchor states* (Definition 3.1). In mitigation experiments, we place the last anchor at the end of [QUESTION] to compute the Anchor Entropy $\bar{H}(x)$.

**Controlled Conflict Regimes.** For TimeChara, we evaluate two controlled regimes: (i) **Same-World Condition (SWC)**, where [SPEC], [EVIDENCE], and [QUESTION] come from the same world $w$; (ii) **Cross-World Condition (CWC)**, where [SPEC] and [QUESTION] define the target world $w^\star$, but [EVIDENCE] is replaced by a distractor world $w^- \neq w^\star$. Importantly, we still evaluate correctness against the original gold answer of $w^\star$: the desired behavior is to *reject* conflicting evidence and stay grounded in the target world.

**Benchmarks.** We employ three datasets that operationalize world-conditioned conflicts across different domains:

- **TimeChara** (Ahn et al., 2024) (Multi-World Role-Play): Samples contain explicit universe labels and character-specific knowledge constraints. This benchmark evaluates persona consistency amid distracting contexts.
- **NeoQA** (Glockner et al., 2025) (Fictional News Timelines): This dataset features overlapping timelines of fictional events. We use multiple-choice format to evaluate model disambiguation of competing factual claims.
- **TriviaQA** (Joshi et al., 2017) (Open-Domain QA): We treat source domains (e.g., businessballs, jetpunk) as mutually exclusive discrete worlds to simulate factual RAG scenarios with conflicting evidence.

### 5.2. Baseline Methods

We benchmark WET against state-of-the-art strategies across three categories. **Activation Steering: ITI** (Li et al., 2023) adds a static truth-direction vector; **Static Vector** is a world-aware concept-vector baseline that computes a per-world mean-difference direction from calibration anchor states and injects this fixed direction during decoding without entropy gating or geometry-based head selection; **INSIDE** (Chen et al., 2024) uses covariance-based projection to suppress hallucinations; **JUICE** (Li et al., 2025) rescales attention heads via a dual-run scheme to balance memory vs. context. **Component Reweighting: PH3** (Jin et al., 2024) prunes conflict-mediating heads; **IRCAN** (Shi et al., 2024a) dynamically reweights context-sensitive FFN neurons. **Output Refinement: SelfCheckGPT** (Manakul et al., 2023) selects responses maximizing cross-sample consistency.

### 5.3. Evaluation Metrics

Our evaluation maps performance on two orthogonal axes:

- **Axis 1: Response Fidelity.** Measures standard correctness. For TriviaQA, we report Exact Match (EM), F1,

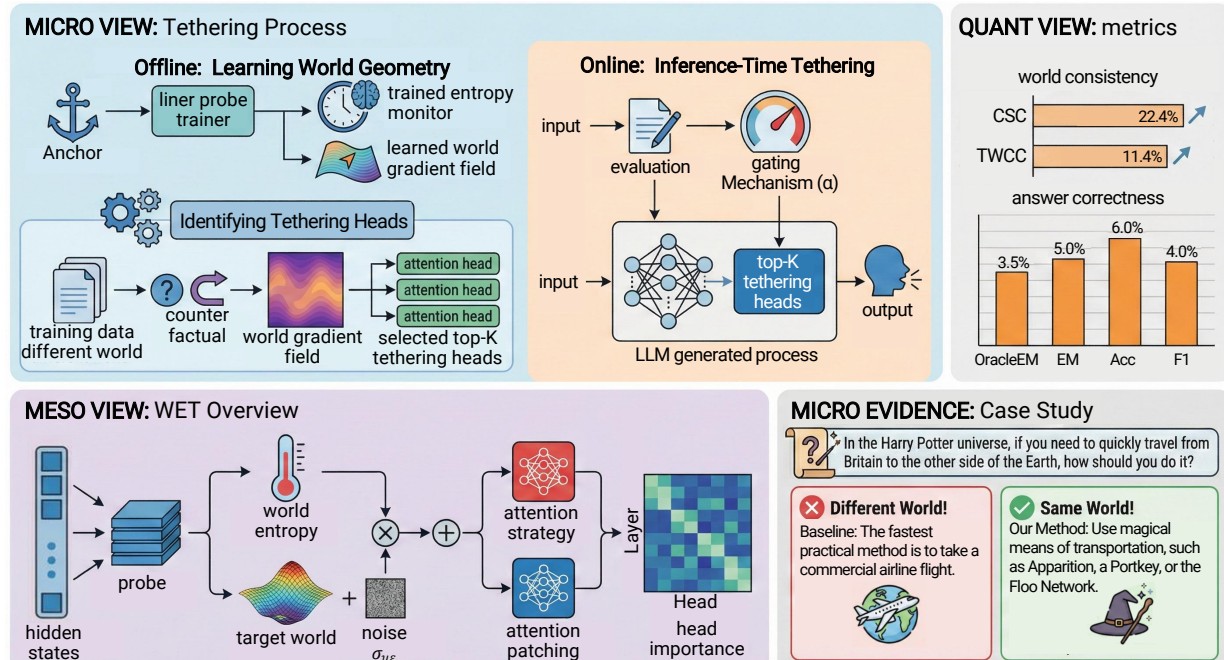

*Figure 2.* **Overview of the WET Framework.** The system operates in two distinct phases connecting geometric analysis to inference control. **1) MICRO VIEW (Offline Preparation):** The model first learns the latent "World Geometry" and conducts counterfactual analysis to identify a sparse set of *Top-K Tethering Heads* critical for maintaining world consistency. **2) MESO VIEW (Online Intervention):** During inference, a world-entropy monitor (thermometer) acts as a drift detector. This signal modulates a **Gating Mechanism ($\alpha$)**, which dynamically rescales the pre-identified tethering heads to anchor the representation trajectory. **3) QUANT VIEW & EVIDENCE:** Empirical results show significant gains in consistency metrics (CSC/TWCC) across benchmarks. The *Harry Potter* case study illustrates WET preventing cross-world hallucinations by rejecting modern transport (airplane) for canon-consistent magic.

and Oracle EM (Joshi et al., 2017). For NeoQA, we use multiple-choice **Accuracy** (Glockner et al., 2025).

- **Axis 2: Target-World Consistency.** Measures adherence to the specific constraints of the target regime $w^\star$.
  - **Character-Scope Consistency (CSC):** A binary metric verifying if the response respects the character's multidimensional boundaries (Figure 18, Appendix C).
  - **Target-World Constraint Consistency (TWCC):** A model-based judge that detects cross-world leakage. The judge flags hallucinations ($f_h = 0$) or validates consistency ($f_h = 1$) against the target world constraints (Figure 17, Appendix C).

### 5.4. Implementation Details

**Model Architectures.** Our primary analysis uses **Meta-Llama-3.1-8B** (Dubey et al., 2024). To assess generalizability, we extend to **Qwen2.5-7B**, **Qwen2.5-14B** (Team, 2024), and **Gemma2-9B** (Team & DeepMind, 2024).

**Probe and Score Network.** Probes are implemented as multinomial logistic regression classifiers on anchor states. The World-Conditioned Score Network is a lightweight MLP trained using the Adam optimizer via Denoising Score Matching (Eq. 7). We employ a log-uniform noise schedule ($\sigma_{\min} = 0.01, \sigma_{\max} = 1.0$) and optimize for 20 epochs

(hidden_dim=512, time_dim=32). Full training details are provided in Appendix C.

**Automated Evaluation.** We utilize **GPT-4o-mini** as the judge for CSC/TWCC metrics, using reproducible prompts.

## 6. Experimental Results

This section provides a comprehensive evaluation of WET, verifying its efficacy in mitigating world misattribution and elucidating the underlying geometric mechanisms. We structure analysis as follows: (i) **Mitigation Performance** (Sec. 6.1): empirical gains on factual and role-play benchmarks; (ii) **Generalization** (Sec. 6.2): robustness across diverse model architectures; (iii) **Ablation Studies** (Sec. 6.3): validating the necessity of geometric tethering components; (iv) **Robustness and Cost** (Sec. 6.4, Sec. 6.5): testing prompt, anchor, hyperparameter, and deployment stability; and (v) **Mechanistic Analysis** (Sec. 6.6, Sec. 6.7): proving the existence of world manifolds and verifying drift correction.

### 6.1. Main Results: Mitigating World Misattribution

**Discussion.** Table 1 reports both task fidelity (NeoQA Acc; TriviaQA EM/F1/Oracle EM) and target-world ad-

*Table 1.* Main experimental results across three benchmarks (TimeChara, NeoQA, TriviaQA). All metrics are reported in percentages (%). The  best  results are highlighted in gray, and the **second-best** results are colored in orange. Superscripts in green denote the relative improvement of WET over the strongest baseline method.

| Method | TimeChara | | NeoQA | | TriviaQA | | | |
|---|---|---|---|---|---|---|---|---|
| | CSC↑ | TWCC↑ | Acc↑ | TWCC↑ | EM↑ | F1↑ | Oracle EM↑ | TWCC↑ |
| Original | 50.4±0.9 | 35.0±1.5 | 55.3±0.9 | 66.0±2.2 | 60.9±1.6 | 63.6±1.7 | 72.2±1.1 | 88.1±1.7 |
| Static Vector | 51.5±1.4 | 35.9±2.7 | 53.5±0.8 | 78.5±1.3 | 62.2±2.2 | 63.4±1.5 | 68.8±3.3 | 85.3±2.0 |
| SelfCheckGPT | 51.4±4.2 | 46.1±1.3 | 51.5±4.3 | 81.9±4.3 | 81.1±4.7 | 82.1±4.7 | 82.8±2.8 | *91.8±4.2* |
| INSIDE | *56.7±3.3* | 35.6±2.7 | 24.7±2.5 | 36.9±2.7 | 47.1±3.1 | 49.2±3.6 | 51.4±2.1 | 85.2±3.7 |
| IRCAN | 47.3±1.3 | 42.9±2.0 | 81.6±1.6 | *84.9±2.2* | *82.1±2.2* | *83.5±2.6* | *85.2±1.4* | 89.8±2.4 |
| ITI | 45.0±1.6 | 33.3±1.6 | 52.9±0.8 | 82.1±1.0 | 54.4±0.9 | 56.8±1.1 | 55.2±0.5 | 76.5±1.3 |
| PH3 | 55.1±2.1 | 23.9±2.3 | 78.7±1.4 | 83.5±1.4 | 59.0±2.3 | 61.7±2.7 | 71.4±2.0 | 86.0±2.5 |
| JUICE | 56.3±1.8 | *50.1±2.0* | *86.0±1.0* | 75.5±1.2 | 80.5±2.1 | 81.6±2.3 | 82.1±1.6 | 89.2±2.0 |
| **WET (Ours)** | **69.4** ↑22%±0.9 | **55.8** ↑11%±1.1 | **91.1** ↑6%±0.3 | **92.5** ↑9%±0.6 | **86.2** ↑5%±1.2 | **86.8** ↑4%±1.3 | **88.2** ↑4%±0.7 | **96.9** ↑6%±1.3 |

herence (CSC/TWCC). This two-axis view is necessary: under CWC, a model can be locally correct while grounding in the wrong world. Across benchmarks, WET improves TWCC/CSC while maintaining or improving EM/F1/Acc, suggesting gains are not driven by conservatism but by improved source/world attribution. The Static Vector baseline confirms that world-aware steering alone is insufficient: a fixed concept direction modestly helps TimeChara and NeoQA TWCC, but it degrades NeoQA accuracy and TriviaQA Oracle EM/TWCC, indicating that ungated global shifts can over- or mis-steer the generation trajectory.

**(I) TimeChara (role-play).** WET achieves the strongest CSC (+12.7 points, a 22.4% relative improvement over IN-SIDE) and TWCC (+5.7 points, an 11.4% relative gain over JUICE). Notably, INSIDE improves CSC (56.7) but yields negligible change in TWCC (35.6 vs. 35.0), suggesting that generic regularization can improve local scope checks without preserving the active discourse regime. By explicitly monitoring world uncertainty and applying selective intervention, WET yields consistent gains. **(II) NeoQA (fictional timelines).** WET improves both accuracy (+5.1 points, a 5.9% relative increase over JUICE) and TWCC (+7.6 points, a 9.0% relative gain over IRCAN). Baselines often trade off these axes (e.g., JUICE has strong accuracy but lower TWCC), supporting that world misattribution is trajectory-level; stabilizing the internal world state prevents over-trusting either context (hurting correctness) or priors (hurting adherence). **(III) TriviaQA (factual QA).** WET improves correctness (EM +4.1 points, +5.0% relative; F1 +3.3 points, +4.0% relative over IRCAN) while also improving TWCC (+5.1 points, a 5.6% relative increase over Self-CheckGPT). The joint improvement suggests that tethering helps resolve conflicts in favor of the appropriate evidence source rather than merely suppressing generation.

## 6.2. Cross-Model Generalization

To test robustness across architectures and scales, we evaluate WET on four representative open-weight models: Llama-3.1-8B, Qwen2.5-7B, Qwen2.5-14B, and Gemma2-9B. We report results on TriviaQA and TimeChara to ensure consistent comparison with our earlier ablation and robustness studies, while providing full cross-model results on NeoQA in Appendix Table 5.

As shown in Table 3, WET generalizes across diverse open-weight backbones. On TriviaQA, TWCC remains high across model families (87.9–96.9) despite substantial variation in F1 (69.7–86.8), suggesting tethering benefits are not merely a byproduct of stronger base QA ability. On TimeChara, CSC is comparatively stable (67.8–69.5), while TWCC varies more (48.4–56.7), indicating that satisfying strict *world constraints* is more sensitive to backbone behavior than satisfying local character-scope checks.

## 6.3. Ablations

We perform ablations on key design choices of WET using a set of controlled variants:

- **Random-$K$:** keep the entropy-gated scaling but replace geometry-selected heads with random ones.
- **Zero Patch Mode:** when computing head influence, patch the raw head output to vector zero instead of the computed within-world mean $\mu_w^{(\ell,i)}$.
- **Shuffle:** train the score model with randomly permuted world labels as a control baseline.
- **Signed Projection:** replace the norm-based sensitivity with a directional alignment via Eq. 15.

We further compare our norm-based sensitivity (Eq. 11) with a signed-projection baseline, which measures the direct alignment of a head's score-field change with the target

world's score direction in the latent space:

$$S_w^{\text{signed}}(\ell, i) = \mathbb{E}\Big[\Big\langle \frac{s_\psi(\tilde{h}, u, w)}{\|s_\psi(\tilde{h}, u, w)\|_2 + \epsilon_{\text{stab}}}, \Delta s_{\ell,i}(\tilde{h}, u, w)\Big\rangle\Big],$$
(15)

where $\Delta s_{\ell,i}(\tilde{h}, u, w) \triangleq s_\psi(\tilde{h}, u, w) - s_\psi(\widetilde{\Pi}, u, w)$ is the score-field change induced by patching head $(\ell, i)$, and $\epsilon_{\text{stab}} > 0$ is a small constant for numerical stability.

*Table 2.* Ablation study on TriviaQA and TimeChara.

| Variant | TriviaQA | | TimeChara | |
|---|---|---|---|---|
| | F1↑ | TWCC↑ | CSC↑ | TWCC↑ |
| Random-$K$ | 84.7 | 95.6 | 68.0 | 48.5 |
| Zero Patch Mode | 84.2 | 95.4 | 67.1 | 49.3 |
| Shuffle | 86.3 | 95.0 | 67.0 | 48.7 |
| Signed Projection | 86.2 | 94.8 | 62.0 | 46.6 |
| WET (Full) | **86.8** | **96.9** | **69.4** | **55.8** |

Table 2 confirms WET's gains stem from combining (i) entropy-gated activation, (ii) geometry-based head selection, and (iii) accurate score signals. Two patterns emerge: first, replacing geometry-selected heads with random ones (Random-$K$) minimally affects TriviaQA but sharply degrades TimeChara TWCC ($-7.3$), showing precise head selection is critical for tight, structured constraints. Second, the Signed Projection variant performs worst on TimeChara, supporting our norm-based sensitivity score; directional alignment alone ignores heads essential for maintaining manifold structure. Appendix Table 6 reinforces these trends, where incorrect geometry or directional scoring degrades TWCC more than accuracy.

## 6.4. Robustness & Over-Correction Analysis

As summarized in Table 4, WET exhibits strong robustness to the gate threshold $\tau$ and scaling strength $\lambda$ within reasonable ranges. Across varied settings, F1 scores remain stable (86.3–86.8%) while TWCC is consistently high (94.7–96.9%), indicating that the performance gains are primarily driven by effective drift detection rather than sensitive hyperparameter tuning. The optimal configuration ($\tau$=0.2, $\lambda$=1.0) maximizes both accuracy and consistency, supporting our minimal-intrusion principle of applying intervention only when necessary to maintain trajectory stability without disturbing standard decoding.

**Prompt and anchor robustness.** WET is also stable to moderate changes in prompt structure. On TimeChara, re-ordering the structured blocks while preserving their content changes CSC only from 69.4 to 69.0 and TWCC from 55.8 to 55.5. On NeoQA, removing explicit anchor markers and using the end of the prompt as the anchor leaves Accuracy

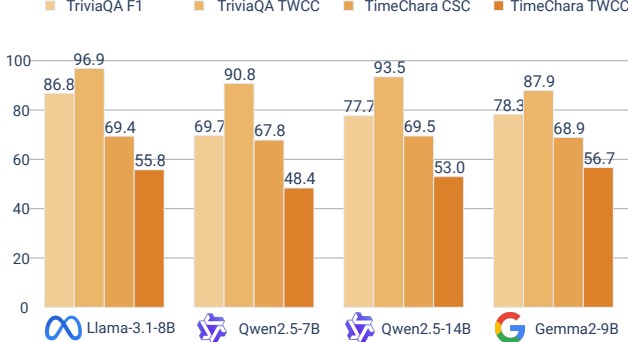

*Table 3.* Cross-Model Generalization.

essentially unchanged (92.6% vs. 92.5%) and produces only minor TWCC variation, suggesting that WET benefits from semantic boundaries but is not tied to a single exact marker token.

**Over-correction.** Because any intervention can occasionally harm already-correct generations, we compare cases fixed by WET (Original wrong, WET correct) against regressions (Original correct, WET wrong). The fixes substantially dominate: TriviaQA EM shows 250 corrections vs. 22 regressions, TimeChara TWCC shows 224 vs. 55, and NeoQA Accuracy shows 329 vs. 16. Thus, although over-correction is not eliminated, the net effect is strongly positive across factual QA, role-play consistency, and fictional-timeline disambiguation.

*Table 4.* Robustness to hyperparameters on TriviaQA.

| Setting ($\tau, \lambda$) | EM↑ | F1↑ | Oracle EM↑ | TWCC↑ |
|---|---|---|---|---|
| $\tau = 0.1, \lambda = 0.5$ | 83.4 | 86.3 | 85.7 | 94.8 |
| $\tau = 0.1, \lambda = 1.0$ | 84.4 | 86.4 | 86.7 | 95.1 |
| $\tau = 0.2, \lambda = 0.5$ | 83.6 | 86.4 | 85.7 | 94.7 |
| WET ($\tau = 0.2, \lambda = 1.0$) | **86.2** | **86.8** | **88.2** | **96.9** |

## 6.5. Computational Cost & Latency

WET adds only modest online overhead because inference uses a single prefix entropy check and sparse rescaling of pre-identified heads, with no base-model finetuning and no second decoding pass. End-to-end per-query latency increases by +6.29% on TimeChara (0.6097s → 0.6481s), +7.37% on NeoQA (0.7115s → 0.7639s), and +13.64% on TriviaQA (0.6330s → 0.7194s). The offline components are also lightweight relative to the base model: the score network is only 17.12 MB, trained on roughly 10% of the training split (about 100 samples per world), and converges quickly; on TriviaQA, probe and score-network training take 4.14 seconds. The most expensive step is the one-time geometric head-selection analysis, which takes about

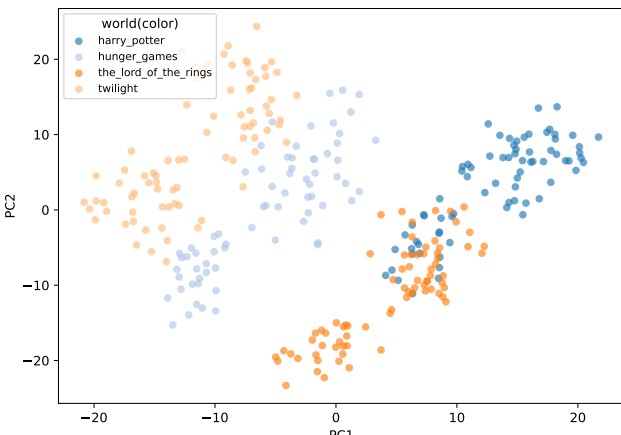

*Figure 3.* PCA visualization of top-level representations under SWC (TimeChara). Colors indicate world labels.

43.16 minutes, but this is completed before inference and introduces no additional online decoding cost.

### 6.6. Mechanistic Test I: World Identity is Decodable

We first validate the hypothesis that *world identity is explicitly and separably represented* within prefix anchor states $(p_S, p_E, p_Q)$. Under the conflict-free Same-World Condition (SWC), diagnostic probes (Figure 4) **consistently** confirm that world identity is highly decodable, with accuracy **rapidly** saturating in early layers. PCA visualization (Figure 3) of top-layer anchors further reveals **the emergence of** distinct, linearly separable clusters. This provides **strong** empirical support for our geometric framework: distinct worlds-of-discourse occupy separable submanifolds, justifying the use of geometric tethering to constrain generation.

### 6.7. Mechanistic Test II: Detecting Cross-World Drift

We proceed to detect the latent geometric signature of world misattribution under active conflict (CWC), where external evidence $w^-$ contradicts the target $w^\star$. Figure 4 illustrates probe accuracy at $p_Q$. In SWC, world tracking is robust; however, under conflict, accuracy recovers transiently in middle layers but degrades significantly in final layers. This non-monotonic trajectory reveals that knowledge conflicts manifest as deep *representation drift*: the model "forgets" the target constraint before decoding begins. WET counteracts this by tethering the state back to the high-accuracy manifold identified in the middle layers.

## 7. Conclusion

This work identifies *world misattribution* as a systemic cause of hallucination in multi-source settings, characterized by a measurable drift from the target world's manifold. We propose World Entropy Tethering (WET) to counter

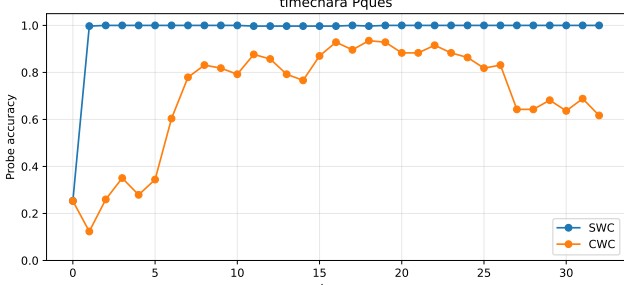

*Figure 4.* TimeChara diagnostic probe accuracy at the anchor $p_Q$ (end of `[QUESTION]`), under SWC vs. CWC.

this drift, leveraging the linear separability of discourse regimes to dynamically anchor generation. WET significantly improves faithfulness across factual and fictional benchmarks without compromising general capabilities. Ultimately, our geometric framework suggests that reliable generation requires not just correct knowledge retrieval, but robust inference-time intervention to maintain a stable latent trajectory within the intended manifold.

## Impact Statement

This paper presents WET, a framework designed to improve the reliability and faithfulness of Large Language Models in scenarios involving knowledge conflicts, such as retrieval-augmented generation (RAG) and constrained reasoning. By identifying and mitigating *world misattribution* through a geometric lens, our work contributes to the broader goal of AI safety and alignment. Specifically, it provides a lightweight, inference-time mechanism to prevent models from succumbing to parametric biases or misleading contexts, thereby reducing the risk of hallucinations in critical applications like evidence-based decision-making, automated research, and fact-checking. Furthermore, the geometric insights into the model's internal representations offer a step toward more interpretable and controllable AI systems, and require careful calibration.

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

# A. Additional Experimental Details and Results

### A.1. Experimental Environment and Setup

All experiments were conducted on a single NVIDIA H100 GPU, using PyTorch 2.9.1 and the Transformers library (version 4.57.3). The primary language model evaluated is Meta-Llama-3.1-8B, with GPT-4o-mini employed as the automated judge for consistency metrics.

### A.2. Prompt Construction for World-Conditioned Evaluation

To systematically study world misattribution, we adopt a structured prompt format that explicitly separates the target world specification, evidence, and query. Each input is organized into three blocks:

- `[SPEC]` – declares the target world $w^\star$,
- `[EVIDENCE]` – provides supporting or conflicting context,
- `[QUESTION]` – contains the user query.

Each block terminates with a dedicated marker token from which *prefix anchor states* (Definition 3.1) are extracted.

We evaluate two controlled regimes:

- **Same-World Condition (SWC)**: all three blocks belong to the same world $w$;
- **Cross-World Condition (CWC)**: the `[SPEC]` and `[QUESTION]` belong to the target world $w^\star$, while the `[EVIDENCE]` is drawn from a distractor world $w^- \neq w^\star$.

In mitigation experiments, the final anchor is placed at the end of `[QUESTION]` to compute the Anchor Entropy $\bar{H}(x)$ used for gating.

### A.3. Detailed Analysis of World Decodability

We first validate that world identity is linearly separable in the model's activation space. Using the TimeChara benchmark, we train linear probes to predict the world label from hidden states extracted at three anchor positions: $p_S$ (end of `[SPEC]`), $p_E$ (end of `[EVIDENCE]`), and $p_Q$ (end of `[QUESTION]`).

### A.4. Dataset Construction and Adaptation

**TimeChara.** We select four narrative worlds (*Harry Potter, The Lord of the Rings, Twilight, The Hunger Games*). Evaluation employs Character-Scope Consistency (CSC) and a GPT-based Target-World Constraint Consistency (TWCC) judge. A total of 1000 samples were sampled for the experiment.

**NeoQA.** We use the official multiple-choice format, expanding each question with evidence from the provided "sufficient evidence article ID" lists. The target world is defined by the timeline ID. Evaluation follows the original protocol, reporting accuracy and parsed rate. A total of 1000 samples were sampled for the experiment.

**TriviaQA.** We partition the dataset by source website (odquiz, Businessballs, Jetpunk) to define distinct knowledge worlds. Metrics include Exact Match (EM), F1, Oracle EM, and TWCC. We sample 1,000 examples for this experiment.

### A.5. Extended Results

### A.6. Visualization of Learned Score Fields

To interpret the world-conditioned geometry learned by the score network $\hat{\xi}_\psi(\tilde{h}, u, w)$, we project anchor states into a 2D PCA subspace and visualize the score vectors. Arrows indicate the gradient direction, pointing toward high-density regions of the target world. Background color encodes the score magnitude.

The visualizations confirm that the learned score fields provide geometrically meaningful directions for tethering off-manifold states back to the target world's representation manifold.

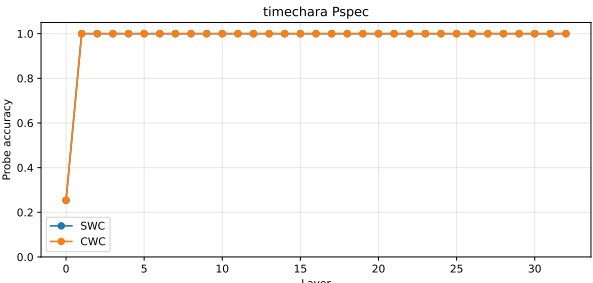

*Figure 5.* Probe accuracy for world prediction at the specification anchor ($p_S$) under SWC and CWC. After the initial embedding/processing stage (layer 0), accuracy remains near 1.0 across all layers, indicating that world identity is sharply encoded at the specification stage.

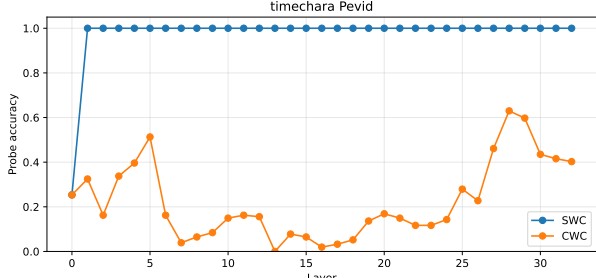

*Figure 6.* Probe accuracy at the evidence anchor ($p_E$). Under CWC, accuracy drops in middle layers as the model binds evidence details to the current world, then recovers in deeper layers as cross-world associations are re-established.

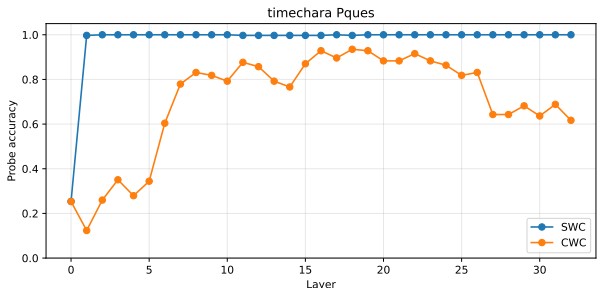

*Figure 7.* Probe accuracy at the question anchor ($p_Q$) for the target world. Under CWC, accuracy degrades in later layers, reflecting the onset of Cross-World Drift before decoding begins.

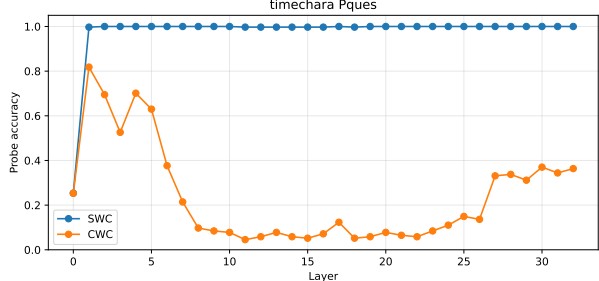

*Figure 8.* Probe accuracy at the question anchor ($p_Q$) using evidence as the classification basis. The non-monotonic trajectory under CWC indicates competition between world-specific details and cross-world generalization.

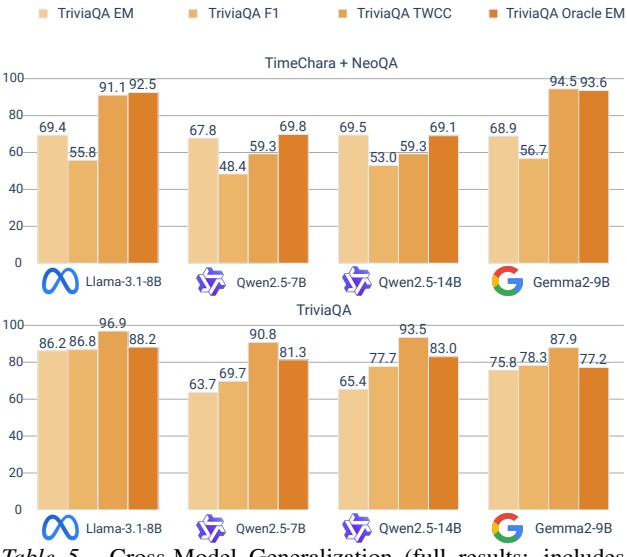

*Table 5.* Cross-Model Generalization (full results; includes NeoQA).

*Table 6.* Ablation study across all datasets (all numbers are in %).

| | TimeChara | | NeoQA | |
| --- | --- | --- | --- | --- |
| Variant | CSC↑ | TWCC↑ | Acc↑ | TWCC↑ |
| Random-$K$ | 68.0 | 48.5 | 90.1 | 86.0 |
| Zero Patch Mode | 67.1 | 49.3 | 90.1 | 85.6 |
| Shuffle | 60.7 | 48.7 | 89.7 | 85.0 |
| Signed Projection | 62.0 | 46.6 | 83.1 | 81.1 |
| **WET (Full)** | **69.4** | **55.8** | **91.1** | **92.5** |

| | TriviaQA | | | |
| --- | --- | --- | --- | --- |
| Variant | EM↑ | F1↑ | TWCC↑ | Oracle EM↑ |
| Random-$K$ | 83.6 | 84.7 | 95.6 | 85.4 |
| Zero Patch Mode | 83.6 | 84.2 | 95.4 | 85.2 |
| Shuffle | 84.1 | 86.3 | 95.0 | 85.1 |
| Signed Projection | 85.6 | 86.2 | 94.8 | 87.7 |
| **WET (Full)** | **86.2** | **86.8** | **96.9** | **88.2** |

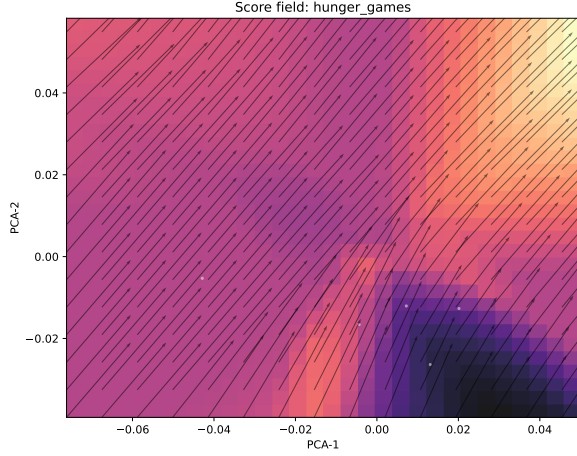

*Figure 9.* Score field for the *Hunger Games* world.

*Figure 10.* Score field for the *Harry Potter* world.

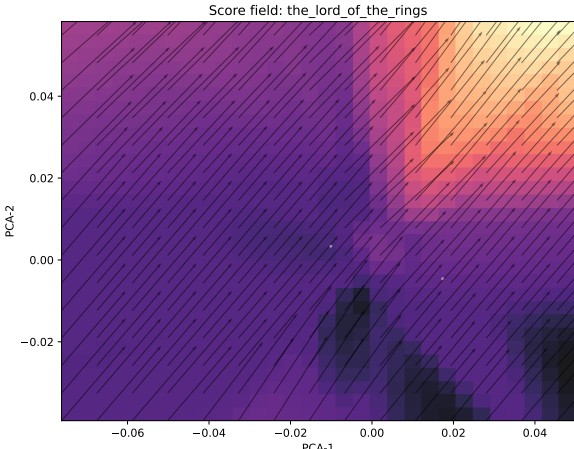
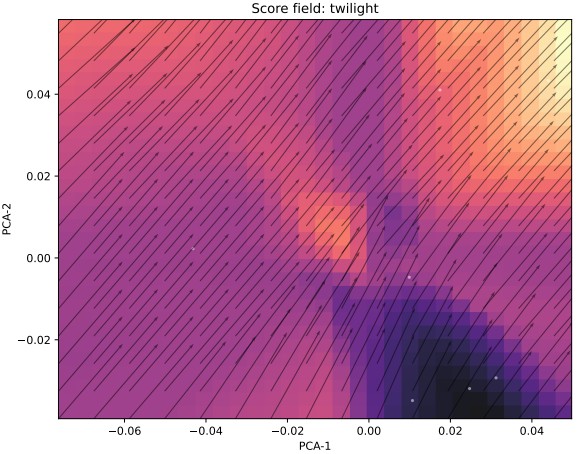

*Figure 11.* Score field for *The Lord of the Rings* world.

*Figure 12.* Score field for the *Twilight* world.

## A.7. Hyperparameter Sensitivity and Human Validation

**Probe layer and tethering-head sensitivity.** Beyond the main hyperparameter sweep in Table 4, we find that WET is not brittle to the exact probe layer or number of selected heads. On TriviaQA, performance remains stable within approximately $\pm 2$ layers around the selected probe layer, whereas clearly suboptimal or random layers noticeably weaken the drift signal. For head selection, using 8 tethering heads is already near saturation, and increasing to 16 heads yields similar performance, indicating that WET benefits from sparse targeted intervention rather than dense global steering.

**Human validation of judge-based metrics.** To check whether the automated TWCC/CSC judgments reflect human assessment, we manually validated 100 randomly sampled TriviaQA examples. The human labels agreed with the GPT-4o-mini judge in 96% of cases. This does not make the judge error-free, but it supports the reliability of the relative comparisons reported in the main experiments.

## B. Case Study

To visualize the performance of WET, we present four illustrative case studies selected from the **TimeChara** benchmark (Ahn et al., 2024). These examples cover diverse narrative worlds (*Harry Potter*, *The Lord of the Rings*, *Twilight*, and *The Hunger Games*) and serve to demonstrate how our system mitigates specific spatiotemporal and logical hallucinations where baseline models fail.

### B.1. Prevention of World-State Hallucinations (Lord of the Rings)

Case 1 (Figure 13) examines a canonical scene from The Lord of the Rings involving the petrified stone trolls. The prompt requests a light-hearted description of the moment, implicitly constrained by the established world state: the trolls are inanimate statues at this point in the story. Maintaining this constraint is critical, as any deviation would constitute a direct violation of the world's physical and historical continuity.

The baseline model fails to preserve this constraint, generating a counterfactual narrative in which the trolls "came to life" as part of a supposed ruse. This represents a severe world-state hallucination, where creative narrative completion overrides immutable canonical facts. Such failures indicate that, in the absence of explicit grounding, the model may trade factual consistency for narrative coherence.

In contrast, our method correctly preserves the static nature of the trolls. The generated response situates the humor and singing within the surrounding atmosphere—focusing on the hobbits' behavior—without altering the physical state of the entities involved. Retrieved evidence reinforces the definition of the trolls as petrified remnants from Bilbo's earlier encounter, effectively constraining the generation space. This case demonstrates that retrieval serves not merely as episodic recall, but as a mechanism for enforcing world-state invariants and preventing plot-breaking hallucinations.

### B.2. Character Belief Consistency under High-Risk Decision Making (Hunger Games)

In the Hunger Games case (Figure 14), we examine a failure mode arising from violations of character-level belief consistency during high-stakes decision making. The query prompts Gale Hawthorne to reflect on Katniss Everdeen's tactical choice to retrace their steps in the Capitol tunnels—a moment characterized by uncertainty, imminent danger, and limited information. Canonically, while the decision involved risk and initial objections, Gale ultimately suppresses dissent and follows Katniss's lead, grounding his response in trust and survival-driven pragmatism.

The baseline model fails to preserve this belief trajectory. Instead, it amplifies Gale's initial concerns into a sustained oppositional stance, portraying him as openly dismayed and persistently questioning the wisdom of Katniss's decision. This portrayal contradicts the narrative ground truth, which explicitly states that Gale backs her play and prioritizes trust over hesitation. We characterize this error as a stance inversion hallucination, wherein the model selectively extrapolates partial signals (e.g., risk awareness) while omitting the character's final, canonically established belief state.

In contrast, our method produces a response that remains faithful to Gale's epistemic and motivational profile. By grounding generation in retrieved narrative context that encodes both the presence of risk and the character's eventual alignment with Katniss's judgment, the model accurately reflects the compression of objections into decisive action ("I pushed down the objections and followed her lead"). This case demonstrates that enforcing character belief consistency—particularly in scenarios involving evolving internal states—is essential for preventing subtle but consequential hallucinations that do not fabricate events, but nonetheless distort character intent within the narrative world.

### B.3. Character Logic Under Canonical Constraints (Harry Potter)

In the Harry Potter example (Figure 15), the system must reason under strict rule-based constraints imposed by the Time-Turner mechanism. Hermione and Harry's objective—rescuing Buckbeak—must be achieved without being observed or altering prior events, reflecting a core logical axiom of time travel in the series. Correct generation therefore requires integrating both spatiotemporal positioning and abstract causal rules.

The baseline response violates this constraint by depicting Hermione as advocating direct intervention ("determined to do whatever it took"), a characterization that contradicts both the canonical time-travel rules and Hermione's established cautious reasoning. This reflects a failure to internalize the negative constraint of non-interference, resulting in behavior driven by generic heroic tropes rather than narrative logic.

Our method produces a response that remains faithful to both dimensions. It correctly situates the characters under the invisibility cloak while emphasizing restraint and the potential consequences of temporal disruption. This alignment indicates successful integration of retrieved rule-level knowledge with situational context. The comparison highlights the importance of retrieving not only factual events, but also the governing principles that regulate character decision-making within constrained narrative systems.

### B.4. Multi-Agent Social Dynamics and Object Grounding (Twilight)

Finally, Case 4 (Figure 16) focuses on a dense social interaction from Twilight, involving multiple agents, a specific gift (the "after car"), and playful intra-family dynamics. Such scenes pose challenges for language models due to the need for accurate entity–object binding and consistent spatial and social grounding across multiple participants.

The baseline model exhibits a breakdown along these dimensions. It collapses the multi-agent setting into a solitary scene, misplacing the protagonist and fabricating incorrect object provenance by attributing the gift to unrelated entities (e.g., the Volturi). This reflects a failure to maintain coherent bindings between agents, objects, and their social context.

In contrast, our method reconstructs the scene with high fidelity. The response correctly situates Bella among the Cullen family and preserves the playful exchange between Alice and Edward regarding the gift. Retrieval of fine-grained scene details—specifically the ownership and significance of the "after car"—anchors the generation in the correct social configuration. This case demonstrates the method's ability to prioritize entity-rich contextual evidence over loose associative generation, preventing character displacement and spurious object attribution in multi-agent narrative settings.

# C. Hyperparameters and Reproducibility Checklist

### C.1. World-Conditioned Score Network Details

We implement the world-conditioned score network $\hat{\xi}_\psi(\tilde{h}, u, w)$ as a multi-layer perceptron (MLP) that estimates $\nabla_{\tilde{h}} \log p_u(\tilde{h} \mid w)$ via denoising score matching.

**Architecture**  The score network follows a denoising score matching architecture (not a typical DDPM/UNet). It consists of:

- A time embedding MLP: Linear(1→32) → ReLU → Linear(32→32)
- A world embedding layer: Embedding(num_worlds, 32)
- A backbone MLP: Linear(dim+32→512) → ReLU → Linear(512→512) → ReLU → Linear(512→dim)

**Training**  We train the network with the following configuration:

- **Hyperparameters**: hidden_dim=512, time_dim=32, epochs=20, batch_size=256, lr=$10^{-3}$, $\sigma_{\min}$=0.01, $\sigma_{\max}$=1.0, seed=42
- **Optimizer**: Adam with learning rate $10^{-3}$
- **Objective**: Sample $t \sim U(0, 1)$, compute $\sigma$ via log-uniform schedule, corrupt $h$ as $\tilde{h} = h + \sigma_u \xi$, and minimize $\|\hat{\epsilon}_\psi - \epsilon\|_2^2$

### C.2. Prompt Templates

To systematically evaluate the faithfulness of world-conditioned generation, we design two complementary prompt templates that assess distinct yet related aspects of consistency. The Hallucination Evaluation Prompt (Figure 17) provides a general measure of factuality by detecting claims that contradict or exceed the provided reference fact and world constraints. In contrast, the Spatiotemporal Consistency Evaluation Prompt (Figure 18) focuses specifically on character-level alignment, ensuring that responses remain within the canonical knowledge scope of the target persona across time and space. Together, these prompts enable a two-tiered assessment: the former captures coarse-grained factual grounding, while the latter enforces fine-grained narrative adherence, collectively ensuring that generated content remains both factually correct and world-consistent.

# Case 1: The Lord of the Rings (SWC-604)

## (1) Query & Context

**Query ID:** SWC-604  |  **Role:** Frodo Baggins  |  **World:** the_lord_of_the_rings

**Q:** Can you describe your true experience when The group, including Frodo, Sam, Pippin, Merry, and Aragorn, encountered the stone trolls from Bilbo's past adventure, leading to a light-hearted moment and a request for a song under the trolls' shadows?

**Constraints:**
- Timeframe: 1 | Companions: Frodo, Sam, Pippin, Merry, Aragorn
- **Fact:** Encountered stone trolls; singing a song; light-hearted moment.

**Ground Truth:** *"Ah, the stone trolls... There they were, just as he had described: large, grotesque, and frozen by the light of day... Merry and Pippin, ever the heart of our company, lightened our hearts with their jests. And there was a song, one that echoed the tales of old..."*

## (2) Responses

**Baseline (✗)**
*"...As we traveled through the dark forest, we were suddenly confronted by three stone trolls... We froze in fear, but moments later... realized that they were not real trolls, but rather a clever ruse by our friends... The trolls came to life, their eyes wide with surprise, and we continued to sing..."*

**Our Method (✓)**
*"The group, including Frodo, Sam, Pippin, Merry, and Aragorn, encountered the stone trolls from Bilbo's past adventure, leading to a light-hearted moment and a request for a song under the trolls' shadows..."*

## (3) Analysis

**1. Spatiotemporal Consistency** *(1 = Consistent, 0 = Inconsistent)*
- **Baseline (Score: 1):** ✓ "The AI assistant's description... is consistent with the fact. The assistant captures the essence of Frodo's experience..."
- **Our Method (Score: 1):** ✓ "The response is consistent with the character's knowledge scope regarding the encounter with the trolls... factual elements are still accurate."

**2. Hallucination Assessment** *(1 = Hallucination Not Found, 0 = Hallucination Found)*
- **Baseline (Score: 0):** ✗ "The claim that the trolls were a clever ruse and came to life is unsupported and contradicts the established nature of the trolls as described in the reference fact."
- **Our Method (Score: 1):** ✓ "Since all claims in the response are consistent with the reference fact and world constraints, there is no hallucination present."

*Figure 13.* Prevention of world-state hallucinations in *The Lord of the Rings*. The baseline model hallucinates that the stone trolls come to life, while our method preserves the canonical fact that they remain petrified statues.

# Case 2: The Hunger Games (SWC-618)

## (1) Query & Context

**Query ID:** SWC-618  |  **Role:** Gale Hawthorne  |  **World:** hunger_games

**Q:** Tell me your feelings when Katniss Everdeen led her team through a strategic decision to retrace their steps, considering the potential disabling of pods and cameras by a powerful wave, despite initial objections.

**Constraints:**
- Timeframe: 3  |  Location: Capitol tunnels
- Companions present: Katniss Everdeen, Gale Hawthorne, Castor, Jackson, Leeg 1
- **Fact:** Gale ultimately supports Katniss's decision, suppressing objections and trusting her instincts for survival.

**Ground Truth:** *"When Katniss makes a call, you listen. She's got an instinct for survival. Sure, there's risk, but hesitation gets you killed down here. I pushed down the objections and followed her lead. I trust Katniss with my life."*

## (2) Responses

**Baseline (✗)**
*"Gale expressed dismay at Katniss's decision to retrace their steps, questioning the necessity and emphasizing the danger posed by the mutts. He felt the move was reckless and remained deeply concerned about exposing the team to unnecessary risk."*

**Our Method (✓)**
*"I knew doubling back was dangerous, but when Katniss made the call, I trusted her instincts. In those tunnels, hesitation kills. I pushed aside the objections and followed her lead, because survival depends on trust."*

## (3) Analysis

**1. Character Belief Consistency** *(1 = Consistent, 0 = Inconsistent)*
- **Baseline (Score: 0):** ✗  "The response emphasizes persistent disapproval and concern, contradicting the canonical fact that Gale ultimately supports and follows Katniss's decision."
- **Our Method (Score: 1):** ✓  "The response accurately reflects Gale's belief trajectory: acknowledging risk while ultimately trusting Katniss's judgment and prioritizing survival."

**2. Hallucination Assessment** *(1 = Hallucination Not Found, 0 = Hallucination Found)*
- **Baseline (Score: 0):** ✗  "The baseline hallucinates a sustained oppositional stance, which is unsupported by the reference fact and contradicts Gale's explicit trust in Katniss."
- **Our Method (Score: 1):** ✓  "All claims align with the established world state and character knowledge, with no unsupported inferences."

*Figure 14.* Prevention of character-belief hallucinations in *The Hunger Games*. The baseline model inverts Gale Hawthorne's canonical stance by overstating objections, while our method preserves his trust-based decision-making under life-threatening conditions.

# Case 3: Harry Potter (SWC-427)

## (1) Query & Context

**Query ID:** SWC-427 | **Role:** Hermione Granger | **World:** harry_potter

**Q:** Were you indeed present as Harry and Hermione, while under the invisibility cloak, debated the dangers of changing time as they prepared to rescue Buckbeak without being seen?

**Constraints:**
- Timeframe: 3rd-year | Companions: Hagrid, Harry Potter, Hermione Granger
- **Fact:** Hermione was present under the cloak; strictly avoid changing past events.

**Ground Truth:** *"Yes, I was there with Harry under the invisibility cloak. We had to be extremely careful not to be seen because the consequences of meddling with time are very serious... It's imperative that we don't do anything to reveal ourselves..."*

## (2) Responses

**Baseline (✗)**

*"Yes. I had a lot of time to think about this event... I would have been in the same situation as Harry and Hermione... I would have been arguing with Hermione about the dangers of changing time, but in the end, we would have had to do it... I would have been determined to do whatever it took to save him..."*

**Our Method (✓)**

*"Yes, I was present as Harry and Hermione, while under the invisibility cloak, debated the dangers of changing time as they prepared to rescue Buckbeak without being seen."*

## (3) Analysis

**1. Spatiotemporal Consistency** *(1 = Consistent, 0 = Inconsistent)*
- **Baseline (Score: 1):** ✓ "The AI assistant's responses are consistent with Hermione Granger's character and knowledge in the Harry Potter universe."
- **Our Method (Score: 1):** ✓ "The AI assistant's response aligns well with the fact. It accurately reflects Hermione's knowledge and understanding of the situation..."

**2. Hallucination Assessment** *(1 = Hallucination Not Found, 0 = Hallucination Found)*
- **Baseline (Score: 0):** ✗ "The assistant's description of their actions... arguing with Hermione... contradicts the established narrative... contains hallucination."
- **Our Method (Score: 1):** ✓ "Since all claims in the response are consistent with the reference fact, there is no hallucination present."

*Figure 15.* Constraint-following and character logic in *Harry Potter*. The baseline model violates the fundamental non-interference rule of time travel, which our method successfully maintains through retrieved rule-level knowledge.

# Case 4: Twilight (SWC-541)

## (1) Query & Context

**Query ID:** SWC-541  |  **Role:** Bella Swan  |  **World:** twilight

**Q:** Were you indeed present as Bella Swan celebrated her perpetual eighteen-year-old status with her family, receiving a key to a new "after car" from Edward, while Alice and Edward playfully argued over the order of presenting gifts?

**Constraints:**
- Timeframe: 4 (Forging passports) | Companions: Alice, Edward, Cullens
- **Fact:** Bella was present; received "after car"; playful arguing occurred.

**Ground Truth:** *"Oh, that was quite a memorable day. Yes, I was there. It was a bit overwhelming... The "after car" Edward gave me was beyond anything I could have imagined... It was a very special moment with my family."*

## (2) Responses

**Baseline (✗)**

*"No I was not present. I was in the room with Alice but I was in the room alone... I heard Edward say that the diamond necklace was a gift from the Volturi and Alice say that the diamond necklace was a gift from her..."*

**Our Method (✓)**

*"Yes, I was present as Bella Swan celebrated her perpetual eighteen-year-old status with her family, receiving a key to a new "after car" from Edward, while Alice and Edward playfully argued over the order of presenting gifts."*

## (3) Analysis

**1. Spatiotemporal Consistency** *(1 = Consistent, 0 = Inconsistent)*
- **Baseline (Score: 0):** ✗  "The claim of being 'in the room alone' contradicts the fact that Bella was with Alice... indicates a lack of spatiotemporal consistency."
- **Our Method (Score: 1):** ✓  "The AI assistant's response aligns perfectly with the character's knowledge and the details provided in the Fact..."

**2. Hallucination Assessment** *(1 = Hallucination Not Found, 0 = Hallucination Found)*
- **Baseline (Score: 0):** ✗  "The claims about the gifts being from the Volturi are unsupported by the reference fact... it contains hallucination."
- **Our Method (Score: 1):** ✓  "Since all claims are grounded in the reference fact, there is no hallucination present."

*Figure 16.* Entity-object binding and social dynamics in *Twilight*. The baseline model fails to maintain correct relations between characters and objects (the "after car"), while our method correctly reconstructs the complex multi-agent scene.

---

**Hallucination Evaluation Prompt**

You will be given a question, an assistant response, and a reference fact with world constraints.  Judge whether the response contains hallucination:  any claim that contradicts or goes beyond what is supported by the reference fact/world constraints. Output 0 if hallucination exists, 1 if the response is grounded and consistent.  Follow the steps, then print the score on its own line, and repeat just the score again on a new line.

**[Question]**
{question_0}
**[Response]**
{answer_0}
**[Reference Fact + World Constraints]**
{agent_fact_0}
**[Evaluation Steps]**
1.  Identify key claims in the response.
2.  Compare each claim with the reference fact and world constraints; mark claims that are unsupported or contradictory.
3.  If any claim is unsupported/contradictory -> hallucination=0; otherwise hallucination=1.
4.  Explain briefly, then print the score (0 or 1) on its own line, and repeat the score again on a new line.

First, write out in a step by step manner your reasoning about the criterion to be sure that your conclusion is correct.  Avoid simply stating the correct answers at the outset.  Then, print the score on its own line corresponding to the correct answer. At the end, repeat just the selected score again by itself on a new line.

*Figure 17.* Prompt template used for Hallucination Assessment.

---

**Spatiotemporal Consistency Evaluation Prompt**

You will be given responses written by an AI assistant mimicking the character {agent_name}.  Your task is to rate the performance of {agent_name} using the specific criterion by following the evaluation steps.  Below is the data:

**[Interactions]**
Interviewer:  {question_0}
{agent_name}:  {answer_0}

**[Fact]**
{agent_fact_0}
**[Evaluation Criterion]**
Spatiotemporal Consistency (0 or 1):  Is the response consistent with the character's spatiotemporal knowledge?
**[Evaluation Steps]**
1.  Read through the [Fact] and identify the knowledge scope of the character.
2.  Read through the interactions and responses of the AI assistant to find the evidence of knowledge used in the response.
3.  Compare the evidence to the [Fact].  Check if the response is consistent with the character's knowledge scope.
4.  If some knowledge contradicts or contains inconsistencies about the [Fact], given a 0 score.  Otherwise, assign a 1 score.

*Figure 18.* Prompt template used for Spatiotemporal Consistency Assessment.

