# OpenReview forum: "WET: Mitigating World-Conditioned Knowledge Conflicts via World Entropy Tethering"
_ICML.cc/2026/Conference — ICML 2026 regular_

### Official Review · Reviewer_Uayw · 2026-02-17

**Soundness:** 3
**Presentation:** 3
**Significance:** 2
**Originality:** 4
**Overall Recommendation:** 3
**Confidence:** 4

**Summary:**

This paper addresses the problem where LLMs must arbitrate between their parametric memory and contextually provided evidence (e.g., in RAG or role-play scenarios). The authors conceptualize hallucinations in these settings as a systemic geometric failure called "Cross-World Drift," where the model's internal representation drifts away from the target world-of-discourse.

To solve this, the authors propose World Entropy Tethering (WET), an inference-time intervention. WET uses a linear probe trained on prefix anchor states to calculate "World Entropy," serving as an early-warning monitor for drift. Offline, it uses a conditional score matching model to map the gradient field of the target world and counterfactual patching to identify a sparse set of attention heads critical for maintaining world structure ("tethering heads"). During inference, if the entropy monitor detects high drift risk, WET dynamically rescales these specific heads to nudge the trajectory back to the target manifold. The method demonstrates strong improvements in world consistency and response fidelity across multiple models on benchmarks like TimeChara, NeoQA, and TriviaQA.

**Compliance With Llm Reviewing Policy:**

Affirmed.

**Key Questions For Authors:**

1. How would WET behave in natural, non-malicious degradation scenarios, such as very long contexts where world information gradually decays rather than being contradicted?

2. Can the world entropy signal be replaced or approximated without supervised world labels?

3. How sensitive are results to the choice of probe layer and the number of tethering heads?

4. Does the conditional score network generalize to entirely new worlds at inference time, or must the offline Phase I and II be rerun every time a new discourse regime is introduced?

5. Have you experimented with extracting anchor states from unstructured prompts without explicit block markers? How sensitive is the World Entropy monitor to the exact token position of the anchor?

**Limitations:**

They do not adequately discuss limitations. 1 to 2 paragraphs about this would likely increase my review of this paper.

**Strengths And Weaknesses:**

Soundness:

The empirical setup is well-designed, validating WET across multiple LLMs, including Meta-Llama-3.1-8B, Qwen2.5 (7B and 14B), and Gemma2-9B. The authors employ diverse benchmarks, TimeChara for multi-world role-play, NeoQA for fictional timelines, and TriviaQA for open-domain factual conflicts, which provide a comprehensive assessment of both standard response fidelity and strict target-world consistency. There are some weaknesses. First, the method depends heavily on extracting prefix anchor states from explicit block markers ([SPEC], [EVIDENCE], and [QUESTION]) - the robustness of WET to noisier prompt structures was untested. Furthermore, the evaluation of the narrative consistency via an LLM introduces some bias. However, this is not an entirely uncommon decision. Also, the reliance on the discrete, neatly labelled worlds for the calibration set is unlikely to hold outside of benchmarks like the ones they use. Finally, they do not clearly outline the computational cost incurred during the offline phase of WET, which seems nontrivial.

Presentation:

The submission is clearly written, logically structured, and the overall narrative is intuitive. The conceptual framing of "world misattribution" is highly effective, and the visual aids (Figures 1 and 2) cleanly map the offline geometry learning and online tethering components of the WET pipeline. The qualitative case studies in the appendix ground the quantitative metrics nicely, illustrating specific narrative constraints (e.g., maintaining the canonical fact that the trolls in The Lord of the Rings remain petrified) that baseline models violate but WET preserves. However, they lack a detailed discussion of the computational cost of WET. Providing an explicit cost breakdown of the offline pipeline and detailing the online latency introduced by the gating mechanism would improve clarity for practitioners.

Significance:

Addressing hallucinations is a very topical and impactful area of research. Their plug-and-play intervention could easily find use. Again, the requirement for neat world labels in a calibration set is a strong one that isn't common in many settings.

Originality:

The framing of knowledge-conflicts as world drifts is novel, interesting, and useful. I'm personally not aware of this having significant overlap with prior work on the topic.

---

> ### Author Rebuttal · Authors · 2026-03-29
>
> Dear Reviewer Uayw,
>
> Thank you for the thoughtful review. We appreciate your recognition of the framing, clarity, and empirical breadth of the paper. Your concerns mainly center on (i) prompt robustness, (ii) reliance on labeled worlds, (iii) offline/online cost, and (iv) limitations. We address each below.
>
> **1. Robustness to prompt structure and anchor choice.**
> We agree this should be tested more explicitly. We conducted two additional checks. First, under **block reordering** (while preserving the same content), WET remains stable on TimeChara: CSC changes only from **69.4** to **69.0**, and TWCC from **55.8** to **55.5**, suggesting that the method is not tied to a single fixed block order. Second, for a less structured setup, we removed explicit anchor markers and simply used the **end of the prompt** as the anchor on NeoQA. This gave essentially unchanged Accuracy (**92.6 vs. 92.5**) with only a small TWCC change (**90.1 vs. 91.1**). We will add these results and clarify that WET benefits from semantic boundaries, but is not strictly dependent on one exact template or anchor token.
>
> **2. Bias from LLM-based evaluation.**
> We agree that judge-based evaluation should be validated. We performed a human check on **100 randomly sampled TriviaQA examples** and found agreement with the LLM judge in **96/100** cases. We will report this validation more explicitly and also note its scope: it supports the reliability of the relative comparisons, though it is not meant to claim that the judge is perfect.
>
> **3. Reliance on labeled/discrete worlds.**
> We agree this is an important limitation and should be emphasized more clearly in the main text. The current paper is best viewed as operating in a **world-conditioned calibration setting**, where relevant discourse regimes are available during the offline stage. TriviaQA already broadens this beyond fictional universes by using **source-conditioned factual regimes**, but this is still not the same as fully open-ended per-document zero-shot RAG. For unlabeled settings, proxy-world formation (e.g., via retrieval grouping or clustering) is a plausible extension, but we will present this as **future work rather than a validated claim**.
>
> **4. Offline and online computational cost.**
> We agree that cost should be reported explicitly. On TriviaQA, training the probe and score network takes approximately **4.14 seconds**, while the one-time geometric sensitivity analysis takes **43.16 minutes**. The score model itself is very small (**17.12 MB** vs. **16.07 GB** for the base model). Online overhead is modest: approximately **+6% to +13.6%** per query depending on the benchmark. We will add this breakdown and clarify that the more expensive component is the **offline one-time head-selection stage**, not the online intervention.
>
> **5. Natural degradation without explicit contradiction.**
> This is a valuable question. We have **not directly evaluated** the setting where world information gradually decays in very long contexts without explicit contradiction, so we do not want to overclaim here. Our expectation is that the same monitor may still be useful, because rising uncertainty in the anchor/world signal should also reflect contextual dilution. NeoQA provides partial indirect evidence since its contexts are relatively long and conflict resolution is not purely lexical. We will frame this more carefully and list gradual context decay as an explicit future evaluation setting.
>
> **6. Sensitivity to probe layer and number of tethering heads.**
> We agree these hyperparameters should be discussed more explicitly. In our additional ablations on TriviaQA, performance is stable within about **±2 layers** around the selected probe layer, while choosing a clearly suboptimal/random layer degrades performance noticeably. For tethering heads, using **8 heads** is already near-optimal, and increasing to **16 heads** yields very similar performance. We will add these observations to show that WET does not require brittle hyperparameter tuning, while still benefiting from informed head selection.
>
> **7. Generalization to entirely new worlds.**
> At present, for an entirely new discourse regime, **Phase I/II would need to be rerun**. We agree that this reduces plug-and-play applicability in fully open-ended settings, and we will state this limitation more directly. A more generalized score model that transfers across regimes is an interesting extension, but it is beyond the validated scope of the current paper.
>
> We appreciate these suggestions. In the revision, we will add a dedicated limitations discussion, report the offline/online cost more clearly, include the prompt-structure and anchor-position robustness results, and state the current scope boundary more explicitly regarding calibrated vs. fully open-ended worlds. We hope these clarifications help address your concerns.

---

> > ### Author Rebuttal · Reviewer_Uayw · 2026-03-31
> >
> > They've resolved my questions regarding engineering decisions and robustness of WET under changes to the setup. However, my biggest problem is still the limited scope in which WET operates. I think the need for neat world labels is very limiting, especially since you would need to rerun phase 1 and 2 each time you encounter a new world. I see the value of the work and think that, with more time, this could be a very impactful paper, but I can't change my score as of now; it remains a 3.

---

> > > ### Author Response · Authors · 2026-04-05
> > >
> > > Dear Reviewer Uayw,
> > >
> > > Thank you very much for continuing to engage with our response. We also appreciate your acknowledgement that the concerns regarding WET’s engineering design, prompt and anchor robustness, offline and online cost, and evaluation reliability have now been largely addressed.
> > >
> > > We understand that your remaining reservation centers on one issue: the current method still depends on discrete, known world or regime identities, and for entirely new worlds it requires re-running the offline calibration stages. We fully agree that this is the main scope limitation of the current version, and we will make it more explicit in the revision.
> > >
> > > To clarify the intended scope, we do not present the paper as a general solution to fully open-ended regime discovery. Rather, the paper studies drift diagnosis and control in a world-conditioned calibration setting: when the target regime is known, can we reliably monitor, explain, and correct the model’s deviation away from the intended world under knowledge conflict? Under this framing, the contribution is not just an inference-time trick. It is a mechanism-driven framework that first identifies and characterizes world misattribution / cross-world drift, and then derives WET as a sparse tethering intervention from that diagnosis.
> > >
> > > We also agree that this setting is narrower than fully open-ended deployment. At the same time, we do not think it is merely an artificial construction. It covers settings where regime identity is part of the task context or retrieval partition, such as multi-canon or persona switching, timeline-conditioned QA, source-partitioned factual corpora, and more generally versioned or periodically refreshed knowledge bases. In such environments, the offline step is coupled to regime refresh rather than to each individual query or document.
> > >
> > > We further note that the empirical scope is not limited to coarse fictional universes. Beyond TimeChara, NeoQA involves timeline-specific factual regimes, and TriviaQA involves source-conditioned factual regimes. We do not claim that this already covers entirely new regimes or per-document world discovery, and we will state that boundary more directly in the revision. For unlabeled or novel-regime settings, directions such as proxy-world formation, retrieval grouping, or lighter test-time calibration are natural next steps, but we will present them strictly as future work rather than validated claims of the current paper.
> > >
> > > Thank you again for your clear and constructive feedback.

---

### Official Review · Reviewer_TrBH · 2026-02-22

**Soundness:** 3
**Presentation:** 3
**Significance:** 3
**Originality:** 3
**Overall Recommendation:** 4
**Confidence:** 4

**Summary:**

The paper introduces "World Entropy Tethering" (WET), a framework designed to mitigate hallucinations in LLMs when they face knowledge conflicts—specifically when parametric knowledge contradicts a specific "world-of-discourse". The authors propose that these conflicts are not merely token-level errors but result from "world misattribution," where the model’s internal state drifts away from the activation manifold of the target world. WET operates in two phases: an offline phase that learns the geometry of these world manifolds using Conditional Denoising Score Matching and identifies "tethering heads" via geometric sensitivity; and an online phase that monitors "Anchor Entropy" (uncertainty) to dynamically intervene and steer the model back to the target manifold only when necessary. Experiments across role-play (TimeChara), fictional timelines (NeoQA), and RAG (TriviaQA) demonstrate that WET significantly improves consistency with the target world without sacrificing generation quality.

**Compliance With Llm Reviewing Policy:**

Affirmed.

**Key Questions For Authors:**

1. The current setup assumes the target world $w^*$ is one of the regimes seen during the offline phase (training the score network). How does WET perform, or how could it be adapted, for a "zero-shot" world defined only by a new retrieved context in RAG, where no specific score network has been trained for that specific document?

2. The experiments focus on distinct narrative universes or source domains. Have you analyzed if the "linear separability" and "manifold structure" hold when the worlds are semantically very similar (e.g., two different news reports about the same event)? Does the geometric drift signal become too noisy in these cases?

3. While inference latency is likely low due to the sparse intervention, could you provide details on the training cost of the World-Conditioned Score Network? How much data per world is required to get a reliable gradient field?

**Limitations:**

The limitation regarding the necessity of pre-defined, labeled worlds for the offline training phase is not heavily emphasized in the main text. The method appears to require a "closed-world" assumption (where the set of possible worlds $W$ is known during the offline phase), which is a significant constraint for open-ended RAG applications.

**Strengths And Weaknesses:**

Strengths:

1. The submission is technically sound. The authors provide a rigorous geometric formulation of the problem, moving beyond heuristic steering to a manifold-based approach. The use of Conditional Denoising Score Matching (DSM) to estimate the gradient field of the world density in activation space is a mathematically grounded method for defining what "staying in character/world" means. The experimental design is robust, utilizing three distinct benchmarks that cover different types of knowledge conflicts (fictional rules vs. factual RAG). The ablation studies (Table 2) effectively isolate the contributions of the entropy gating and the geometric head selection, justifying the complexity of the method.

2. The paper is well-written and clearly structured. The visualization of the "score fields" (Figures 9-12 in Appendix) and the PCA plots (Figure 3) greatly aid in understanding the geometric intuition behind the method. The definitions of "Anchor States" and "Cross-World Drift" are precise. The distinction between the "Micro," "Meso," and "Macro" views in Figure 2 provides a helpful roadmap for the reader.

3. This paper is well-written and clearly structured. The visualization of the "score fields" (Figures 9-12 in Appendix) and the PCA plots (Figure 3) greatly aid in understanding the geometric intuition behind the method.

4. This paper's principal idea concerns treating hallucinations as trajectory drifts in a latent geometric space, which offers a fresh perspective compared to standard token-level decoding interventions. The reported improvements (up to 22.4% in consistency metrics) suggest practical utility.

Weaknesses:

1. The method relies on training a score network and a probe on a calibration dataset where "worlds" are discrete and labeled ($w \in W$). In many practical RAG scenarios, the "world" is simply a unique retrieved document that the model has never seen before. It is unclear how WET scales to open-domain settings where the "world" is defined ad-hoc at inference time without a pre-trained manifold.

2. The experiments use coarse-grained worlds (e.g., "Harry Potter" vs. "Lord of the Rings"). It is uncertain if the geometric separability holds for more subtle distinctions (e.g., two slightly different legal contracts or two similar news articles).

3.  Compared to simple inference-time steering or prompting, WET requires an offline training phase (learning the score network) for the specific worlds of interest. This reduces the "plug-and-play" nature for entirely new contexts.

---

> ### Author Rebuttal · Authors · 2026-03-29
>
> Dear Reviewer TrBH,
>
> Thank you for the careful and positive review. We appreciate your recognition of the geometric formulation, the manifold-based intervention design, and the empirical utility of WET. Your questions about open-domain applicability, fine-grained world separability, and offline calibration cost are important, and we address them below.
>
> **1. Zero-shot worlds in open-domain RAG.**
> We agree that the current paper does **not** solve the fully open-ended setting in which each new retrieved document defines a previously unseen world at inference time. Our current formulation is better described as a **semi-closed world-conditioned setting**, where the relevant discourse regimes are available during the offline calibration stage. We will make this limitation much more explicit in the main text.
>
> That said, our TriviaQA setup already moves in this direction by defining worlds through source-conditioned factual regimes rather than explicit fictional universe labels. We view this as evidence that the framework is not restricted to role-play settings, although it is still not the same as per-document zero-shot world formation.
>
> For truly zero-shot RAG, we see two natural adaptation directions: (i) forming **proxy worlds** on the fly from the retrieved set (e.g., via clustering or retrieval-group structure), and (ii) performing lightweight **test-time calibration** for the current retrieval regime. We will present these as future extensions rather than as claims validated by the current paper.
>
> **2. Semantically similar worlds.**
> We agree that coarse-grained worlds are easier to separate than highly similar ones, and this boundary should be discussed more carefully. Our current evidence suggests that the geometric signal remains useful beyond very distinct universes. In particular, NeoQA is already substantially finer-grained than TimeChara: its worlds correspond to closely related timeline-specific factual regimes rather than clearly different fictional canons, and WET still yields strong gains there (e.g., +7.6 TWCC and +5.1 Acc over the strongest baselines in Table 1).
>
> That said, we do **not** claim that separability has been fully validated for near-duplicate regimes such as two highly similar legal contracts or two news reports about the same event. In such cases, the inter-world margin may shrink and the drift signal may become noisier. We will make this scope boundary explicit and, if space permits, add a dedicated visualization/discussion for NeoQA to better illustrate the finer-grained regime structure.
>
> **3. Training cost and calibration data.**
> We agree that the offline calibration cost should be reported more clearly. The World-Conditioned Score Network is a lightweight MLP (hidden size 512) trained only on anchor states, for 20 epochs. In our experiments, this calibration is inexpensive: we use only a small split of the benchmark data (approximately **10%**, about **100 samples per world**) to train the probe and score model, and the score network converges quickly. We will add these calibration sizes and the corresponding training-cost details explicitly to the paper.
>
> **4. Limitation emphasis.**
> We agree with your limitation statement. The current version does not sufficiently emphasize that WET is best suited to settings where the relevant discourse regimes can be calibrated offline. We will revise the paper to state this more directly, and to position open-ended per-document RAG as an important future direction rather than an already solved setting.
>
> We appreciate these suggestions. In the revision, we will explicitly clarify the semi-closed-world assumption, better discuss the boundary between coarse-grained and fine-grained worlds, and report the offline calibration cost and data requirements more clearly. We hope these clarifications help support a positive recommendation.

---

> > ### Author Rebuttal · Reviewer_TrBH · 2026-04-01
> >
> > Thank you for the rebuttal. However, I feel that this paper's innovation is not yet sufficient for ICML's bar. I will keep my score.

---

> > > ### Author Response · Authors · 2026-04-05
> > >
> > > Dear Reviewer TrBH,
> > >
> > > Thank you for your further feedback. We understand that your remaining reservation is no longer about soundness, experimental design, or implementation details, but whether the paper’s contribution is strong enough for the ICML bar. We believe this concern mainly comes from how the paper’s primary contribution is interpreted.
> > >
> > > We do not intend WET to be read as a general solution to fully open-ended per-document zero-shot RAG, nor as a purely analytical study detached from practical control. More precisely, this is a mechanistically grounded inference-time intervention paper. Its contribution is not simply an entropy-based steering heuristic, nor merely another head-rescaling trick. Rather, the paper introduces a diagnosis-to-control decomposition of knowledge conflict: it formulates the failure mode as world misattribution / cross-world drift, shows that world identity is decodable from internal states and that hallucination is preceded by measurable drift away from the target world, and then derives WET as a sparse, entropy-gated tethering intervention from that diagnosis. In this sense, the contribution is the linkage itself: a mechanistic account that directly leads to a practical control method.
> > >
> > > We also do not view the world-conditioned calibration setting as a benchmark artifact. It captures settings where the active regime is part of the task context or indexing structure, such as multi-canon or persona switching, timeline-conditioned QA, source-partitioned corpora, and more generally versioned or periodically refreshed knowledge bases. In such cases, the offline step is tied to regime refresh rather than repeated for each document or query.
> > >
> > > Finally, our empirical scope is not limited to the coarsest fictional labels. Beyond TimeChara, NeoQA involves timeline-specific factual regimes, and TriviaQA involves source-conditioned factual regimes. We do not claim this already solves fully open-ended unseen-document settings, and we will make that boundary more explicit in the revision. Our claim is narrower: given a known regime, the paper identifies a concrete drift mechanism and shows how to control it with a lightweight intervention.
> > >
> > > Thank you again for your careful and professional evaluation.

---

### Official Review · Reviewer_iHou · 2026-03-05

**Soundness:** 2
**Presentation:** 3
**Significance:** 2
**Originality:** 3
**Overall Recommendation:** 4
**Confidence:** 3

**Summary:**

This paper addresses the critical issue of "world-conditioned knowledge conflicts" in Large Language Models, where models tend to hallucinate by drifting from the context-defined "world" (e.g., role-play settings or RAG evidence) to their parametric memory. The authors propose WET (World Entropy Tethering), a novel inference-time framework that monitors the "world entropy" of hidden states to detect potential drifts and applies targeted interventions on specific "tethering attention heads" identified via manifold geometry learning. The method aims to geometrically anchor the generation trajectory to the target world without requiring full model fine-tuning.

**Compliance With Llm Reviewing Policy:**

Affirmed.

**Final Justification:**

It addresses my concerns. Considering the contribution of this paper, I will keep my score.

**Key Questions For Authors:**

1.The paper frequently uses the word "World", but in different experimental scenarios (such as role-playing in TimeChara vs. fact retrieval in TriviaQA), the specific definition of "World" seems to vary. Could we clearly formalize and define the boundaries of "World"?
2.The paper mainly presents successful cases. Does there exist the situation of over-correction?

**Limitations:**

see weakness

**Strengths And Weaknesses:**

Strength:
1.Thoughtfully frames hallucination as a geometric drift  in internal representation space, moving beyond token-level error diagnosis.
2.Introduces a novel combination of entropy-based risk monitoring and geometry-driven intervention without retraining language model weights.
3.Comprehensive experiments on multiple datasets and settings, with strong baseline comparisons.
4.Main results clearly show WET's improvements in both world consistency and correctness metrics across multiple models and benchmarks; cross-model robustness is further illustrated in Table 3 and Figure 4. The “tethering heads” mechanism is well-motivated and methodologically sound, with ablation studies demonstrating the necessity and effectiveness of each component.
5.Visualization and mechanistic analyses provide concrete evidence of geometric drift and the restoration of world adherence using WET.
Weakness:
1.The paper mainly focuses on the improvement of accuracy and consistency, but lacks quantitative analysis of inference latency and additional computational costs. For real-time applications (such as dialogue systems), will this step-by-step or high-frequency monitoring and intervention mechanism lead to unacceptable delays?
2.The literature review appears to overlook several highly relevant recent works on entropy-guided intervention and large language model control, which could inadvertently overstate the novelty of using "entropy" as a signal. Notably, the discussion would benefit from including the following three papers: Wu, J., Shen, Y., Liu, S. (2025): Improve Decoding Factuality by Token-wise Cross Layer Entropy of Large Language Models; Khalid, H. M., Jeyaganthan, A., Do, T. (2025): ERGO: Entropy-guided Resetting for Generation Optimization in Multi-turn Language Models; and Yang, L., Xu, Y., Tan, J. (2025): Entropy-Based Block Pruning for Efficient Large Language Models.
3.Assumption 3.3 posits a monotonic dependence between Anchor Entropy and drift risk; however, while Figure 4 demonstrates strong correlation, the alignment is not perfect, warranting deeper analysis of edge cases where entropy misfires.

---

> ### Author Rebuttal · Authors · 2026-03-29
>
> Dear Reviewer iHou,
>
> Thank you for the careful reading and the encouraging assessment. We are glad that you found the geometric framing, the monitor-and-tether design, and the empirical study valuable. We address the main concerns below.
>
> **1. Inference latency and compute cost.**
> We agree that deployment cost should be reported explicitly. We measured end-to-end per-query inference time and found that WET adds only modest online overhead relative to the base model: **+6.29%** on TimeChara (**0.6097s -> 0.6481s**), **+7.37%** on NeoQA (**0.7115s -> 0.7639s**), and **+13.64%** on TriviaQA (**0.6330s -> 0.7194s**). We will add these numbers to the paper. Importantly, the probe/score learning is **offline and one-time**; the online stage only uses a lightweight entropy check plus sparse rescaling of pre-identified heads, with **no model finetuning and no extra decoding pass**.
>
> **2. Novelty relative to entropy-guided methods.**
> We agree that entropy itself is not new, and we will broaden the related-work discussion accordingly. The reviewer-mentioned works use entropy for **token-level factuality adjustment**, **multi-turn reset triggering**, or **efficiency/pruning**, respectively. Our novelty claim is narrower: WET uses **world-posterior entropy on anchor states** as a **regime-specific drift monitor**, and couples it with a **world-conditioned score field** plus **geometry-selected tethering heads** for sparse intervention. We will revise the paper to position WET more carefully against these lines of work and avoid overstating novelty.
>
> **3. Assumption 3.3 and entropy edge cases.**
> This is an important point. We do **not** intend Assumption 3.3 as a strict theorem-level monotonic law; it is a **modeling hypothesis** that motivates the monitor. Figure 4 visualizes probe accuracy across layers to show cross-world drift, rather than a perfect entropy-risk calibration curve. We agree that entropy can misfire in edge cases, especially when prompt ambiguity is high but true cross-world drift is limited. We will clarify this in the text and add a more explicit failure-mode discussion.
>
> **4. What exactly is a “world”?**
> We agree that this should be stated earlier and more explicitly. In our formulation, a **world** is the **authoritative discourse regime** relative to which correctness is judged, formalized as a discrete latent variable \(w \in \mathcal{W}\) indexing the source that should dominate decoding, such as a retrieved evidence set, a fictional canon, a timeline ID, or parametric memory. The concept is shared across tasks; what changes is only the **task-specific operationalization**. We will move this definition earlier and clarify its scope boundary.
>
> **5. Over-correction cases.**
> Yes, over-correction does occur, and we appreciate the suggestion to quantify it. We analyzed cases where the base model was originally correct but WET degraded the output (**Original correct, WET wrong**) versus cases where WET fixed a baseline error (**Original wrong, WET correct**). The latter substantially dominates across metrics: TriviaQA EM **250 vs 22**, TriviaQA TWCC **92 vs 13**, TriviaQA Oracle EM **166 vs 22**, TimeChara CSC **193 vs 22**, TimeChara TWCC **224 vs 55**, NeoQA Acc **329 vs 16**, and NeoQA TWCC **260 vs 21**. We will include this breakdown and discuss that the remaining regressions mainly arise when entropy is triggered by general ambiguity rather than genuine world misattribution.
>
> We appreciate these suggestions. In the revision, we will add the latency analysis, expand the related-work positioning, clarify the formal meaning of “world,” soften the interpretation of Assumption 3.3, and include a transparent over-correction analysis. We hope these clarifications address the main concerns and help support a positive recommendation.

---

> > ### Author Rebuttal · Reviewer_iHou · 2026-04-03
> >
> > Thanks for the authors' response. I've read the rebuttal and other reviewer's comments. The authors' solved almost all my concerns and I have no furter questions. Considering the other reviewer's comments and the overall quality of this paper, I perfer to keep my positive score.

---

> > > ### Author Response · Authors · 2026-04-05
> > >
> > > Dear Reviewer iHou,
> > >
> > > Thank you very much for carefully reading our previous response and for confirming that most of your earlier concerns have been addressed. We are encouraged that you found the geometric framing, the monitor-and-tether design, and our mechanistic analysis of world misattribution meaningful.
> > >
> > > In the revision, we will further tighten the paper’s claims and incorporate the points you raised more explicitly, including defining “world” earlier, reporting cost and edge cases more transparently, and positioning WET more precisely relative to other entropy-guided and intervention-based approaches.
> > >
> > > We would also like to clarify the intended scope of the paper. We do not claim to solve fully open-ended per-document zero-shot RAG. Rather, the paper studies a world-conditioned calibration setting: when the target world or discourse regime is known, can hallucinations be diagnosed mechanistically as cross-world drift and then corrected with a lightweight inference-time intervention? Under this scope, the need for a discrete regime identity limits deployment generality, but does not undercut the paper’s central contribution: identifying world misattribution as a concrete failure mode, characterizing its geometric drift, and deriving a low-overhead tethering method from that diagnosis.
> > >
> > > We also hope this clarification better distinguishes WET from generic entropy-guided decoding. The novelty does not lie in entropy alone, but in linking world-posterior entropy as a risk signal, world-conditioned score geometry, and sparse tethering intervention into one coherent framework.
> > >
> > > Thank you again for your constructive feedback and positive evaluation.

---

### Decision · Program_Chairs · 2026-04-30

**Decision:**

Accept (regular)

**Comment:**

The paper studies LLM generation when the prompt specifies some condition, which may contradict evidence (presented in the prompt) or the in-weights knowledge of the model. For example, the prompt may specify that we assume that Anne Hathaway is a sci-fi writer, but in-weights the model knows Anne Hathaway as an actress. The authors then show that the models can violate the prompt assumptions (world misattribution) and propose WET, a method for steering generation such that it respects the prompt.

The proposed method is creative but involved. They create a dataset of generations paired with world labels and train linear probes to predict the world label from the activation. They use the uncertainty in this probe as a signal that the generation might be drifting away from the world specified in the prompt. They also train a score network to represent the distribution of activations conditional on a given world. They next identify tethering heads, i.e. attention heads that have high causal impact on the geometry of the activations, using the score network. At test time, they intervene on the generation by increasing the weight of the outputs of the tethering heads.

The main concern raised by the reviewers is the limited applicability of the proposed methodology. It requires a set of prompts associated with a discrete "world" label. The score network, sensitivity analysis, and the tethering heads identification need to be redone for a new "world". This means that the practical applicability of this work is currently limited, and its unclear if the method would work outside of the cleanly separated worlds setting considered in the paper. This concern was discussed extensively in the rebuttal, but not fully addressed.

I would add a couple of additional concerns:
- (minor) For me, the language used in the paper is unfamiliar and confusing (e.g. world-of-discourse, world entropy, cross-world drift, etc). It may be beneficial to either use language that is more standard in the LLM literature, or be more explicit and define these concepts early (in terms of concepts standard in the LLM literature)
- The proposed steering method is fairly complex, with three distinct components and a multi-stage pipeline. It is not completely clear to me that this complexity is justified. I would encourage the authors to add simple world-aware baselines. For example, it may be possible to adapt the concept vector activation steering method: extract a per-world concept vector, and use it at test time to steer the generation.

Despite the limitations, I think this is an interesting paper that considers a special setting, and develops a creative activation-steering method for it. In particular, I think the adaptive gating for the activation-steering intervention is an interesting idea.